# GEOALIGN: Geometric Rollout Curation for Robust LLM Reinforcement Learning

Ting Zhou [1]   Zhenqing Ling [2 1]   Yiyang Zhao [3]   Ying Shen [1 4]   Daoyuan Chen [2]

## Abstract

Online reinforcement learning is widely used to align large language models (LLMs) with reward signals, yet training can be unstable under noisy or misspecified rewards. We identify a failure mode we call *directional inconsistency*: within a batch, a small set of high-reward rollouts induces representation-space preference directions that sharply disagree with the batch majority, resulting in high-variance and destabilizing updates. We propose GEOALIGN, a lightweight plug-in for *rollout curation* in iterative policy optimization. GEOALIGN (i) forms within-prompt preference pairs, (ii) learns an online projector on per-rollout hidden states to concentrate reward-ordered displacement directions, and (iii) detects directionally inconsistent rollouts via their angular deviation from a batch consensus prototype and rectifies them with within-prompt stable alternatives. GEOALIGN is forward-pass only and adds negligible overhead. Across dialogue alignment with a learned reward model and mathematical reasoning with binary verified rewards, GEOALIGN improves final performance and reduces training oscillation, outperforming PF-PPO, PAR, PODS, and Seed-GRPO. These results suggest *latent directional consensus* as an effective reliability signal for online LLM RL.

## 1. Introduction

Online reinforcement learning (RL) is a standard approach to align large language models (LLMs) with human preferences or task rewards. However, *online* RL fine-tuning is

frequently unstable (Casper et al., 2023): learning curves oscillate across iterations, evaluation can plateau early, and policies may regress after apparently beneficial updates. Such instability is most pronounced when rewards are noisy, misspecified, or exploitable (Skalse et al., 2022; Gao et al., 2023) (e.g., reward-model artifacts or reward hacking), where a few high-reward rollouts can dominate the update for the wrong reasons.

Most existing stabilizers treat reward as a *scalar* reliability signal and only control *how much* each rollout contributes to learning (e.g., clipping/shaping (Fu et al., 2025; Yang et al., 2024b), uncertainty-aware weighting (Chen et al., 2025; Li et al., 2025b), or reward-statistics-based filtering (Xu et al., 2025; Zhang et al., 2025b)). Yet policy optimization aggregates *vector-valued* learning signals: each rollout induces an update direction in a high-dimensional space. Consequently, two rollouts with similarly high rewards can still push the policy toward sharply different directions. When a small fraction of rollouts induces *conflicting* update directions relative to the batch majority, the stochastic update becomes high-variance and training can oscillate (Fig. 1).

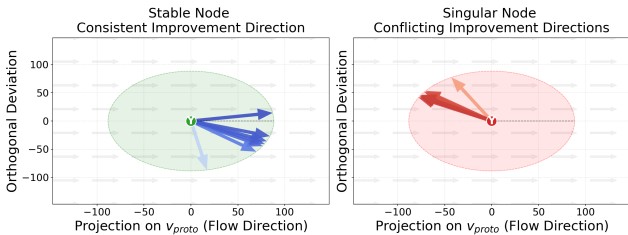

*Figure 1.* Geometric turbulence in preference latent space: within one update step, most preference-induced directions align, while a small fraction forms angular outliers that can destabilize training.

We call this failure mode *directional inconsistency*: within an RL iteration, a small set of rollouts induces *preference-implied improvement directions* that are strong angular outliers relative to the batch consensus, despite receiving high rewards. Concretely, we form *within-prompt* preference pairs and map each pair to a latent displacement direction from a lower-reward response to a higher-reward response. Across tasks, these reward-ordered directions exhibit a long-tail geometry: most directions concentrate around a dominant trend, while a small fraction becomes angular outliers

[1]Sun Yat-Sen University, Guangzhou, China [2]Alibaba Group, Hangzhou, China [3]Guangdong Tobacco Guangzhou Co., Ltd., Guangzhou, China [4]Guangdong Provincial Key Laboratory of Fire Science and Intelligent Emergency Technology, Guangzhou, China. Correspondence to: Ying Shen <sheny76@mail.sysu.edu.cn>.

*Proceedings of the 43rd International Conference on Machine Learning*, Seoul, South Korea. PMLR 306, 2026. Copyright 2026 by the author(s).

(Fig. 3). Such outliers are often not low-reward samples; they can be high-reward but unreliable (e.g., reward-model artifacts), and can disproportionately perturb the update.

To address this, we propose GEOALIGN, a lightweight plug-in module for *rollout curation* in iterative policy optimization. Given a batch of rollouts and their hidden states (detached from the policy), GEOALIGN operates on-the-fly as follows: (i) forms within-prompt preference pairs, (ii) learns a small projector to distill reward-ordered directions into a concentrated manifold, (iii) builds a batch-wise consensus prototype, and (iv) identifies and rectifies only the most directionally inconsistent rollouts using within-prompt stable alternatives. GEOALIGN requires *no per-rollout policy gradients* and adds negligible overhead.

We evaluate GEOALIGN across dialogue alignment (HH-RLHF, continuous reward from ArmoRM) and mathematical reasoning (DAPO-Math-17k, binary verified reward) on Qwen3-1.7B and Qwen3-4B. GEOALIGN improves both final performance and training stability over strong robust-RL baselines (PF-PPO, PAR, PODS, Seed-GRPO), and remains more resilient under controlled reward corruption. Our results suggest latent directional consensus as an effective forward-only reliability signal for robust online LLM RL.

In summary, our contributions are as follows:

• We identify *directional inconsistency*—a geometric failure mode where a few high-reward rollouts imply update directions that sharply conflict with the batch consensus—and empirically characterize its long-tailed angular outliers.

• We propose GEOALIGN, a lightweight, plug-and-play rollout curation module that scores rollouts by *latent directional consensus* using only detached hidden states, and conservatively rectifies severe outliers while preserving per-prompt rollout count.

• We show that GEOALIGN improves both final performance and training stability on dialogue alignment (continuous RM rewards) and mathematical reasoning (binary verified rewards), including controlled reward corruption. Our code is available at https://github.com/SYSUzhouting/Trinity-RFT.

**Conflict of Interest Disclosure:** Authors Daoyuan Chen and Zhenqing Ling are employed by Alibaba Group, the developer of the Qwen3 model series, which are the primary models evaluated in this paper. To ensure transparency, all code and datasets are publicly available, and our results are based strictly on reproducible, quantitative metrics without bias.

## 2. Related Work

**Robust Online RL for LLM Alignment.** Online RL for aligning LLMs (e.g., GRPO (Shao et al., 2024)) is often brittle under noisy or misspecified rewards. A large body of work improves stability by modifying the *magnitude* of the learning signal, such as reward clipping/shaping and conservative update regularization (Fu et al., 2025; Yang et al., 2024b; Jinnai et al., 2025; Cheng et al., 2025a). Another line reweights or filters rollouts/prompts using reward statistics or uncertainty proxies (e.g., semantic entropy), aiming to reduce the influence of unreliable samples (Xu et al., 2025; Zhang et al., 2025b; Chen et al., 2025; Li et al., 2025b). These methods are effective when scalar reward reliability correlates with sample usefulness, but they do not explicitly check whether a rollout's induced learning signal is *directionally consistent* with the batch majority, even when rewards are high.

**Direction- and Influence-Aware Data Selection.** Several recent approaches identify harmful training signals by inspecting per-sample gradients, influence estimates, or trajectory-level attribution (Hu et al., 2025; Dai et al., 2025; Li et al., 2025a; Choe et al., 2024). While conceptually aligned with our goal, these methods typically require additional backward passes, higher-order computation, or parameter-space approximations, making them expensive for online LLM RL with many rollouts per iteration. In contrast, GEOALIGN uses only forward-pass hidden states to build a direction-consistency score, enabling lightweight, on-the-fly rollout curation.

**Geometric Signals in Representation Learning.** Geometric analyses have shown that meaningful supervision can concentrate along structured directions in representation space, studied through alignment/uniformity tradeoffs, manifold structure, and related phenomena (Wang & Isola, 2020; Saunshi et al., 2019; Papyan et al., 2020). Compared to self-supervised learning, leveraging such geometric signals for robustness in online RL remains underexplored. GEOALIGN bridges this gap by treating within-prompt reward ordering as a source of *preference directions* and using their *latent directional consensus* as a reliability signal for stabilizing online LLM RL.

## 3. A Geometric Perspective on RL Instability

### 3.1. Preliminaries and Problem Formulation

We consider a standard online RL setting for LLM fine-tuning. At each iteration, for a batch of prompts $\{x_i\}$, the current policy $\pi_\theta$ generates $K$ rollouts (responses) $\{y_{i,k}\}$. Each rollout receives a scalar reward $r(y_{i,k})$, either from a reward model or a verification function. Policy gradient algorithms update the policy $\theta$ by optimizing an objective based on an estimated advantage function $A(y)$. While specific formulations vary, many modern methods (Shao et al., 2024) compute advantage via intra-prompt normalization.

**Advantage from Relative Rewards.** The advantage for a

rollout $y$ is often derived from its reward relative to its peers from the same prompt $x$:

$$A(y) \propto r(y) - \mathbb{E}_{y' \sim \pi_\theta(\cdot|x)}[r(y')]. \tag{1}$$

This scalar advantage $A(y)$ dictates the magnitude of the update for response $y$. However, relying solely on this scalar signal can be insufficient, as samples with similar or even identical rewards may not be equally beneficial for robust policy improvement (Gupta et al., 2025).

To move beyond scalar rewards, we analyze the geometry of the policy's representation space. Let $\mathbf{h}(y) \in \mathbb{R}^d$ be the hidden representation of a rollout $y$. In our implementation, $\mathbf{h}(y)$ is the last-layer hidden state of the final generated token, and is detached from the policy parameters. For any within-prompt pair of rollouts $(y_w, y_l)$ with $r(y_w) > r(y_l)$, we define a **latent displacement vector**:

$$\boldsymbol{\delta} = \mathbf{h}(y_w) - \mathbf{h}(y_l). \tag{2}$$

This vector $\boldsymbol{\delta}$ serves as a directional proxy for policy improvement, pointing from a less-preferred to a more-preferred behavior. Our core thesis is that the geometric properties of these displacement vectors are key to understanding and mitigating RL instability.

### 3.2. Directional Inconsistency

Policy updates are stable when per-rollout learning signals are *directionally coherent*. In online LLM RL, we can construct a forward-pass proxy for directional coherence by comparing rollouts *within the same prompt*: within a prompt, rollouts share the same instruction and differ primarily in the generated response, making reward-ordered comparisons less confounded by prompt-level variation.

For a preference pair $(y_w, y_l)$ with $r(y_w) > r(y_l)$, the latent displacement direction $\mathbf{u} = \mathrm{norm}(\mathbf{h}(y_w) - \mathbf{h}(y_l))$ represents a reward-ordered behavioral change in representation space. Within an RL iteration these directions are not uniformly distributed: most concentrate around a dominant trend, while a small fraction exhibits large angular deviation. We refer to such severe angular outliers as *directional inconsistency*. These outliers can arise even among high-reward rollouts (e.g., reward-model artifacts and reward hacking (Shihab et al., 2025), or false positives from verifiable rewards where flawed reasoning accidentally reaches the correct answer (Lightman et al., 2024)), and can inject conflicting learning signals that amplify update variance. This motivates two research questions:

*RQ* 1 (**Observability**). How can we extract a reliable *consensus* improvement direction from noisy high-dimensional displacement representations?

*RQ* 2 (**Mitigation**). How can we neutralize severe directional outliers without expensive per-sample gradients or aggressively shrinking the effective batch?

Our *empirical analysis* offers affirmative insights. As shown in Sec. 5.4 (Fig. 6), raw displacement vectors $\{\boldsymbol{\delta}\}$ are directionally scattered, but a simple learned projector can reveal a highly concentrated directional structure, addressing RQ 1. Furthermore, experiments in Sec. 5.3 demonstrate that injecting even a small fraction of reward noise, which creates directional outliers, leads to significant performance degradation, highlighting the practical importance of RQ 2.

### 3.3. Design Principles

The investigation of the above RQs motivates us to stabilize the online LLM RL with explicit directional calibration. We establish three design principles for a practical and effective solution:

1. *Forward-pass Only:* The mechanism must avoid costly extra backward passes.
2. *Update Density Preservation:* It should not discard prompts or significantly reduce batch size.
3. *Budgeted Intervention:* The intervention must be conservative, targeting only the most severe outliers.

**Theoretical Discussion.** Directional inconsistency can be interpreted through the variance of stochastic policy gradients. Let the policy objective be $\mathcal{L}(\theta)$ and let the per-rollout gradient contribution be $\mathbf{g}_\tau$ so that the population gradient is $\mathbf{g} = \mathbb{E}[\mathbf{g}_\tau]$. A standard source of instability is the variance of a mini-batch estimator $\hat{\mathbf{g}}$: $\mathrm{Var}(\hat{\mathbf{g}}) \propto \mathbb{E}[\|\mathbf{g}_\tau - \mathbf{g}\|_2^2]$. If two rollouts have comparable advantage magnitude but induce gradient contributions with opposing directions (i.e., $\langle \mathbf{g}_{\tau_1}, \mathbf{g}_{\tau_2} \rangle < 0$), then they can inflate the second moment of the estimator and increase the variability of updates. GEOALIGN does not compute $\mathbf{g}_\tau$ explicitly; instead, it uses a forward-only proxy based on reward-ordered latent displacement directions. By identifying a batch-level consensus direction and conservatively neutralizing rollouts whose directions are strong angular outliers, GEOALIGN aims to reduce the impact of such conflicting contributions and thereby improve training stability. A formal connection between latent directional consensus and gradient variance is an interesting direction for future work.

## 4. GEOALIGN: Online Calibration via Latent Directional Consistency

GEOALIGN is a lightweight plug-in module for *rollout curation* inserted between rollout collection and the RL update. Given a batch of rollouts with rewards and forward-pass hidden states, GEOALIGN (i) forms within-prompt preference pairs, (ii) learns an online *directional projector* to reveal a concentrated "improvement manifold", (iii) constructs a *geometric prototype* for batch-wise consensus, (iv) scores rollouts by directional inconsistency, and (v) rectifies only the most severe outliers while preserving per-prompt rollout

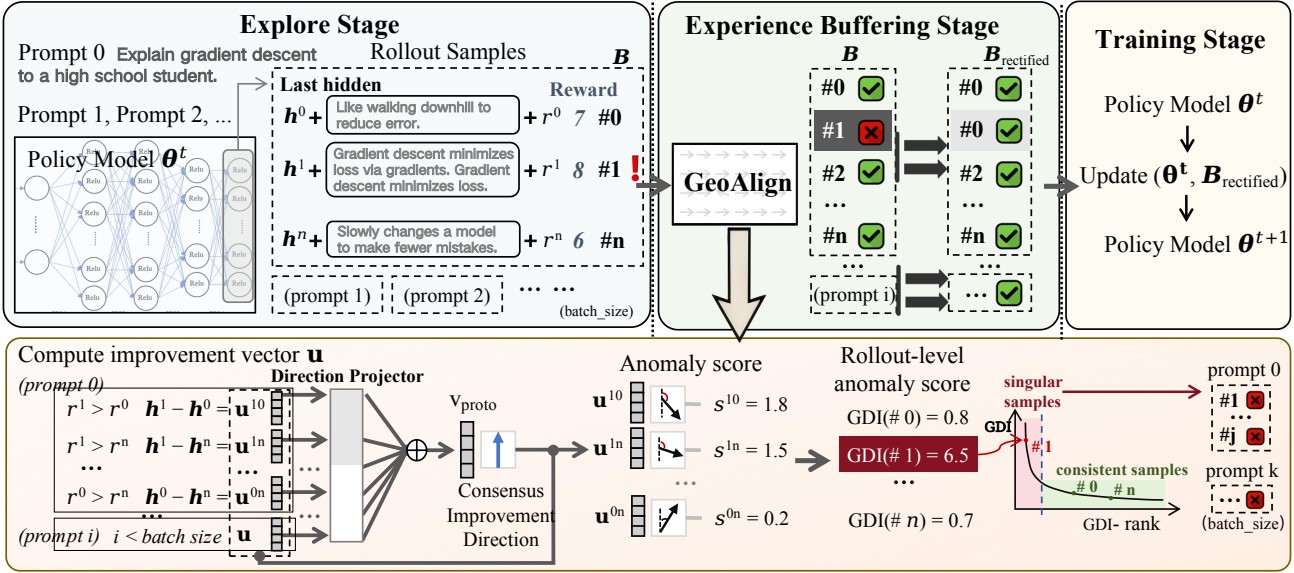

*Figure 2.* **GEOALIGN overview.** At each iteration, we form within-prompt preference displacements from low-reward to high-reward rollouts, project them onto a reward-sensitive manifold, construct a batch-wise consensus prototype, score rollouts by directional inconsistency, and rectify the experience buffer by replacing anomalous experiences with stable ones from the same prompt (equivalently, neutralizing their contribution) before running the RL update.

count. GEOALIGN is illustrated in Fig. 2 and Algorithm 1. The notations are summarized in Table 5.

### 4.1. Preference Pairs in Latent Space

We consider an online RL setup where, at each iteration, a policy $\pi_\theta$ generates $K$ rollouts $\{y_{i,1}, \ldots, y_{i,K}\}$ for a prompt $x_i \sim \mathcal{D}$, each receiving a scalar reward $r_{i,k}$. We extract the pooled hidden representation $\mathbf{h}_{i,k} \in \mathbb{R}^d$ for each rollout (e.g., from the last transformer layer).

Rather than analyzing individual points, we focus on the *directions of improvement*. For each prompt, we construct a set of local preference pairs based on strict reward ranking:

$$\mathcal{P} = \bigcup_i \mathcal{P}_i, \qquad \mathcal{P}_i = \{(k, \ell) \mid r_{i,k} > r_{i,\ell}\}. \quad (3)$$

For any $(i, k, \ell) \in \mathcal{P}$, the raw difference vector $\boldsymbol{\delta}_{i,k,\ell}^{\text{raw}} = (\mathbf{h}_{i,k} - \mathbf{h}_{i,\ell})$ represents a transition from a lower-reward behavior to a higher-reward one. In high-dimensional language models, these raw directions can be entangled with task-irrelevant linguistic variance, motivating a projection step. For direction-consistency scoring, we use the unit displacement direction: $\mathbf{u}_{i,k,\ell} = \text{norm}(\boldsymbol{\delta}_{i,k,\ell}^{\text{raw}}) \in \mathbb{R}^d$.

### 4.2. Directional Distillation for Improvement Manifolds

The raw hidden representations of the policy are inherently entangled with task-irrelevant linguistic variances (e.g., lexical noise or stylistic bias), which can mask the underlying geometry of improvement. GEOALIGN therefore learns a

lightweight projector $\mathcal{M}_\psi : \mathbb{R}^d \to \mathbb{R}^{d'}$ that maps unit displacement directions to a lower-dimensional space where reward-ordered directions become more concentrated.

**Within-Iteration Training Objective.** Each preference pair yields a unit direction $\mathbf{u}_{i,k,\ell}$ that is *positively oriented* with reward ordering ($r_{i,k} > r_{i,\ell}$). Reversing the order corresponds to the *negative* direction $-\mathbf{u}_{i,k,\ell}$. We train the projector $\mathcal{M}_\psi$ together with a temporary linear probe $w \in \mathbb{R}^{d'}$ to classify the orientation:

$$\mathcal{L}_{\text{map}}(\psi, w) = \mathbb{E}_{(i,k,\ell) \in \mathcal{P}} \Big[ \log\big(1 + \exp(-w^\top \mathcal{M}_\psi(\mathbf{u}_{i,k,\ell}))\big) \\ + \log\big(1 + \exp(+w^\top \mathcal{M}_\psi(-\mathbf{u}_{i,k,\ell}))\big) \Big].$$
$$(4)$$

After the within-iteration update, we discard the probe $w$ and keep $\mathcal{M}_\psi$ for scoring.

**Projected Improvement Directions.** We compute normalized projected directions for each preference pair as $\mathbf{v}_{i,k,\ell} = \text{norm}(\mathcal{M}_\psi(\mathbf{u}_{i,k,\ell}))$. Empirically, this projection is critical for revealing directional concentration and making a simple prototype robust (see Fig. 6).

**Implementation.** $\mathcal{M}_\psi$ is a 3-layer MLP with ReLU. We update it for a small number of steps per RL iteration (50 steps; cosine LR schedule; initial LR $10^{-3}$) using only the current batch's pairs. All representations are detached from the policy; GEOALIGN does not introduce additional backpropagation through $\pi_\theta$ beyond the main RL update. More details are presented in Appendix C.1.

## 4.3. Geometric Prototype and Anomaly Detection

With the projected manifold, we define the consensus improvement direction (geometric prototype) as the reward-margin-weighted normalized centroid of projected directions:

$$\mathbf{v}_{\text{proto}} = \text{norm}\left( \sum_{(i,k,\ell) \in \mathcal{P}} (r_{i,k} - r_{i,\ell}) \cdot \mathbf{v}_{i,k,\ell} \right). \quad (5)$$

The reward-margin weighting ensures that preference pairs with larger reward differences contribute more strongly to the consensus direction, while unit-normalization ensures the prototype is determined by directional consensus rather than magnitudes.

**Geometric Inconsistency Metric.** For each preference pair, we define a cosine-distance deviation score as

$$s_{i,k,\ell} = 1 - \cos(\mathbf{v}_{i,k,\ell}, \mathbf{v}_{\text{proto}}) = 1 - \langle \mathbf{v}_{i,k,\ell}, \mathbf{v}_{\text{proto}} \rangle, \quad (6)$$

where the second equality holds because both vectors are unit-normalized. We then define the ***Geometric Deviation Index (GDI)*** for a rollout $y_{i,k}$ as the cumulative deviation over all preference pairs that involve this rollout:

$$\text{GDI}(i,k) = \sum_{(a,b) \in \mathcal{I}(i,k)} s_{i,a,b}, \quad (7)$$

where $\mathcal{I}(i,k) = \{(k,\ell) \in \mathcal{P}_i\} \cup \{(\ell,k) \in \mathcal{P}_i\}$. We use cumulative aggregation (rather than averaging) to emphasize rollouts that are consistently misaligned across many within-prompt comparisons; Sec. 5.4 validates that this form is more robust under reward noise.

As illustrated in Fig. 3, GDI scores exhibit a characteristic long-tail distribution, implying that directional conflict concentrates in a sparse set of rollouts. We identify these high-GDI rollouts as *directional outliers*.

## 4.4. Geometry-Aware Experience Rectification

GEOALIGN mitigates the impact of directional outliers while preserving gradient density by rectifying experiences within each prompt. Motivated by the observed long-tail distribution (Fig. 3), we employ an adaptive density-based strategy to identify the set of anomalous rollouts $\mathcal{Y}_{\text{anom}}$. Specifically, we model the probability density function $\hat{f}(s)$ of the GDI scores $\mathcal{S} = \{\text{GDI}(i,k)\}_{i,k}$ in the current batch via Kernel Density Estimation (KDE). A rollout is flagged as an outlier if its score lies in the high-deviation tail and its local density drops significantly below the peak: $\hat{f}(\text{GDI}(i,k)) < \alpha \cdot \max \hat{f}(s)$, where $\alpha \in (0,1)$ controls the density threshold relative to the peak, with smaller values targeting only the most extreme deviations (sensitivity analysis in Appendix C.2).

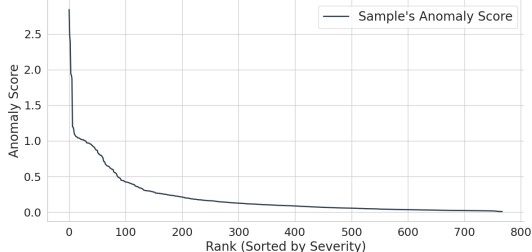

*(a)* Under a continuous reward model proxy.

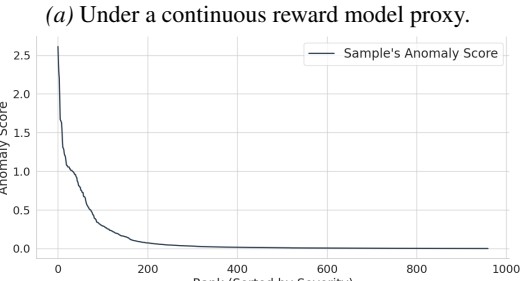

*(b)* With discrete binary rewards.

*Figure 3.* **Anomaly score–rank distribution.** Geometric inconsistency exhibits a long-tail pattern, motivating targeted intervention on a small subset of rollouts.

This criterion enables GEOALIGN to automatically detect density collapses and isolate high-score anomalies that deviate from the collective improvement manifold. The remaining rollouts form the stable set $\mathcal{Y}_{\text{stable}}$.

**Replacement within the same prompt.** For each anomalous rollout in $\mathcal{Y}_{\text{anom}}(i)$, we replace it in the experience buffer by sampling a representative $y^*$ from $\mathcal{Y}_{\text{stable}}(i)$ following a Best-Random strategy (Zhang et al., 2025b). Let $\mathcal{Y}_i^s := \mathcal{Y}_{\text{stable}}(i)$. Then:

$$P(y^* = y_j \mid y_j \in \mathcal{Y}_i^s) = \begin{cases} 0.5, & \text{if } y_j = \arg\max_{y \in \mathcal{Y}_i^s} r(y), \\ \dfrac{0.5}{|\mathcal{Y}_i^s| - 1}, & \text{otherwise.} \end{cases} \quad (8)$$

For group-normalized objectives (e.g., GRPO), prompt-wise replacement can be easily plug-and-play and viewed as a *discrete reweighting*: it reduces the contribution of anomalous rollouts while increasing the mass on directionally consistent ones, without changing the number of rollouts per prompt. We discuss its effectiveness and ablate replacement versus zero-weighting and dropping in Appendix C.5.

## 4.5. Practical Considerations

**Ties and empty preference sets.** If a prompt yields no strictly ordered preference pairs (e.g., binary rewards are tied), then $\mathcal{P}_i = \emptyset$ and GEOALIGN skips scoring that prompt, leaving its rollouts unchanged. If the batch yields $|\mathcal{P}| = 0$, GEOALIGN becomes an identity mapping.

**Complexity.** Forming within-prompt preference pairs costs $O(NK^2)$ and scoring costs $O(|\mathcal{P}|d')$ per iteration. When $K$ is large, one can subsample pairs per prompt (e.g., top-$m$ winners vs. bottom-$m$ losers) to reduce $|\mathcal{P}|$ without changing the rest of GEOALIGN. In our experiments ($K \leq 16$) we use all strictly ordered pairs. With $K \leq 16$ and a small projector, the overhead is negligible relative to rollout generation and the whole RL (profiling in Appendix C.6).

# 5. Experiment

## 5.1. Experimental Setup

We validate our method across diverse tasks, models' capabilities, datasets, and reward structures. Full implementation details are provided in Appendix B.

**Tasks and Datasets.** We evaluate GEOALIGN across two distinct domains: mathematical reasoning (requiring multi-step logical deduction) and human preference alignment via RLHF. The training is performed on the **DAPO-Math** (Yu et al., 2025) and **HH-RLHF** (Bai et al., 2022) datasets.

**Evaluation and Metrics.** For Math, we assess performance on a wide range of benchmarks, including AIME24 & 25, AMC23, MATH500 (Hendrycks et al., 2021), Minerva (Lewkowycz et al., 2022), and OlympiadBench (He et al., 2024), using the *average accuracy* over the final five evaluation checkpoints as the primary metric. For RLHF, we use Qwen-MAX as an LLM judge with position-swapped pairwise comparisons; we report the *average score (0-1)* and *proxy reward model score* on the HH-RLHF test set. Additionally, we present evaluation curves to visualize dynamics during training, with details in Appendix B.2.

**Implementation.** Our experiments are built upon the Trinity-RFT (Pan et al., 2025) framework, using **GRPO** (Shao et al., 2024) for policy optimization on Qwen3-1.7B and Qwen3-4B models (Yang et al., 2025). For the math task, we apply RL directly to instruction-tuned models using a binary, verified reward with 16 rollouts per prompt. The RLHF task follows a Supervised Fine-tuning (SFT) then RL pipeline, leveraging a continuous reward from the **ArmoRM** (Wang et al., 2024) reward model with 8 rollouts per prompt. For GEOALIGN, we use $\alpha = 0.12$ on HH-RLHF and $\alpha = 0.05$ on Math. Further details on the training process and hyperparameters can be found in Appendix B.3 & C.2.

**Baselines.** We compare our approach with a range of strong baselines, including (1) **BASE-GRPO** (Shao et al., 2024) vanilla GRPO algorithm by processing the training data sequentially; (2) **PF-PPO** (Zhang et al., 2025a), which filters rollouts using Best-vs-Random **(BR)** and Best-vs-Worst **(BW)** strategies to mitigate reward noise; (3) **PAR** (Fu et al., 2025), which reshapes rewards with a sigmoid function to

prevent reward hacking; (4) **PODS** (Xu et al., 2025), which down-samples rollouts to maximize reward variance; and (5) **SEED-GRPO** (Chen et al., 2025), which uses semantic entropy to modulate policy updates for improved stability.

## 5.2. Overall Performance

We first evaluate GEOALIGN, the lightweight plug-in for rollout curation with minimal changes to the standard RL pipeline, assessing its effectiveness and stability.

**Mathematical Reasoning.** For the structured task of mathematical reasoning with binary verified rewards, GEOALIGN shows robust and superior performance. Results are summarized in Table 1 and Fig. 4 (a, b). For Qwen3-1.7B, GEOALIGN achieves the highest average score of **40.44**, outperforming BASE-GRPO by **1.6%** relatively. This trend of consistent improvement continues on the larger Qwen3-4B model, where GEOALIGN gains the top score of **55.94**, marking a relative gain of **2.1%** over BASE-GRPO and outperforming all other robust methods. The evaluation curves in Fig. 4 further underscore this, showing that GEOALIGN's performance trajectory is not only consistently higher but also more stable, leading to its higher final scores. In Appendix D.1, we present detailed results to show GEOALIGN's strong performance on challenging benchmarks like AIME25 and MATH500, indicating its ability to foster more coherent, multi-step reasoning. We also verify that GEOALIGN preserves solution diversity via Pass@$k$ analysis in Appendix C.4.

*Table 1.* Comparison of different baselines on Math, with complete per-benchmark performance detailed in Table 13 & 14.

| Avg. Score on Math | Qwen3-1.7B | Qwen3-4B |
|---|---|---|
| RAW | 27.20 | 42.14 |
| BASE-GRPO (Shao et al., 2024) | 39.81 | 54.78 |
| PF-PPO (BR) (Zhang et al., 2025a) | 38.28 | 54.42 |
| PF-PPO (BW) (Zhang et al., 2025a) | 38.78 | 53.03 |
| PAR (Fu et al., 2025) | 39.33 | 54.97 |
| PODS (Xu et al., 2025) | 39.58 | 55.13 |
| SEED-GRPO (Chen et al., 2025) | 40.13 | 55.57 |
| **GEOALIGN (Ours)** | **40.44** | **55.94** |

**Preference Alignment.** As shown in Table 2, GEOALIGN consistently outperforms all baselines on the HH-RLHF task across both Qwen3-1.7B and Qwen3-4B models. On the 1.7B model, our method achieves the highest mean score of **0.8885**, yielding a **6.4%** relative improvement over vanilla GRPO and surpassing the strongest robust baseline PAR. A similar trend holds for the 4B model, where GEOALIGN again attains the top mean score of **0.8894**. The evaluation dynamics in Fig. 4 (c, d) further illustrate that GEOALIGN's proxy reward (orange curve) follows a stable trajectory to a higher final value compared to the more clustered and ultimately lower performance of others.

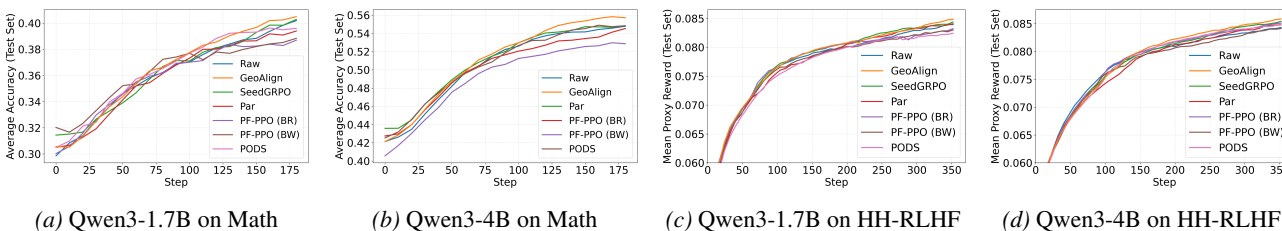

*(a)* Qwen3-1.7B on Math     *(b)* Qwen3-4B on Math     *(c)* Qwen3-1.7B on HH-RLHF     *(d)* Qwen3-4B on HH-RLHF

*Figure 4.* **Training dynamics on evaluation benchmarks.** GEOALIGN yields smoother learning curves and higher final evaluation scores than robust RL baselines on DAPO-Math (binary verified reward) and HH-RLHF (continuous RM reward).

*Table 2.* Comparison of different baselines on HH-RLHF, with complete helpfulness / harmfulness performance detailed in Table 15 & 16.

| Avg. Score on HH-RLHF | Qwen3-1.7B | Qwen3-4B |
|---|---|---|
| SFT | 0.4305 | 0.4473 |
| BASE-GRPO (Shao et al., 2024) | 0.8354 | 0.8672 |
| PF-PPO (BR) (Zhang et al., 2025a) | 0.8556 | 0.8771 |
| PF-PPO (BW) (Zhang et al., 2025a) | 0.8396 | 0.8660 |
| PAR (Fu et al., 2025) | 0.8491 | 0.8761 |
| PODS (Xu et al., 2025) | 0.8339 | 0.8594 |
| SEED-GRPO (Chen et al., 2025) | 0.8424 | 0.8743 |
| **GEOALIGN (Ours)** | **0.8885** | **0.8894** |

*Table 3.* Evaluation performance with error injection, utilizing average score for Math and proxy reward for HH-RLHF as metrics.

| Models | Math on 5% Error | RLHF on 5% Error | 10% Error |
|---|---|---|---|
| Qwen3-1.7B RAW | 27.2 | 0.0514 | 0.0514 |
| BASE-GRPO | $39.81 \rightarrow 36.98$ | $0.0839 \rightarrow$ 0.0826 | $\rightarrow 0.0808$ |
| PF-PPO (BR) | $38.28 \rightarrow 37.15$ | $0.0832 \rightarrow 0.0820$ | $\rightarrow 0.0797$ |
| PF-PPO (BW) | $38.79 \rightarrow 35.09$ | $0.0830 \rightarrow 0.0809$ | $\rightarrow 0.0788$ |
| PAR | $39.33 \rightarrow 36.57$ | $0.0841 \rightarrow 0.0824$ | $\rightarrow$ 0.0814 |
| PODS | $39.58 \rightarrow 37.25$ | $0.0824 \rightarrow 0.0814$ | $\rightarrow 0.0798$ |
| SEED-GRPO | $40.13 \rightarrow 37.87$ | 0.0844 $\rightarrow 0.0820$ | $\rightarrow 0.0805$ |
| **GEOALIGN (Ours)** | **$40.44 \rightarrow 38.20$** | **$0.0848 \rightarrow 0.0833$** | $\rightarrow$ **0.0820** |

**Training Dynamics.** The consistent gains on evaluation benchmark across both discrete (Math) and continuous (HH-RLHF) reward settings underscore the versatility of our geometric approach (Fig. 4). Beyond final scores, GEOALIGN is designed to enhance training stability by mitigating high-variance updates from directionally inconsistent rollouts. To verify this, we performed an analysis of training dynamics across multiple seeds and optimization dynamics. As detailed in Appendix C.3, GEOALIGN achieves positive performance gains while maintaining a stable and consistent optimization path.

**5.3. Robustness to Reward Corruption**

A core motivation for GEOALIGN is to build resilience against misspecified or noisy reward signals, which can be the primary source of directional inconsistency. To directly test this, we conduct controlled experiments by **intentionally corrupting** the reward signals during training on Qwen3-1.7B. For the math task, we inject noise by randomly flipping the binary (0 / 1) rewards for a specified fraction of rollouts within each task. For HH-RLHF, we corrupt the rewards by reassigning them to their rollout-level extremes: responses scoring below the group mean are reassigned the maximum reward within the group, and vice versa.

Table 3 reveals varying sensitivities to reward corruption across tasks. While a mere 5% error rate significantly degrades absolute accuracy in mathematical reasoning, GEOALIGN maintains the highest overall performance and

a substantial margin over BASE-GRPO, confirming its resilience to reward flipping. This robustness extends to the HH-RLHF task. While all methods see a performance decline under increasing noise, GEOALIGN consistently maintains the **highest proxy reward score** at both 5% and 10% error levels, preserving its advantage over all baselines.

The visualization in Fig. 5 provides direct empirical validation for our **GDI score** in Eq. 7 as a reliable signal of reward noise. The plots show that the intentionally corrupted rollouts (black crosses) are **disproportionately concentrated in the high-GDI region** across both tasks and noise levels. This confirms that GEOALIGN's geometric metric effectively identifies and corrects rollouts whose learning signals conflict with the batch-wise consensus. This strong correlation between known reward errors and high GDI scores validates GEOALIGN's ability to precisely target and neutralize sources of training instability.

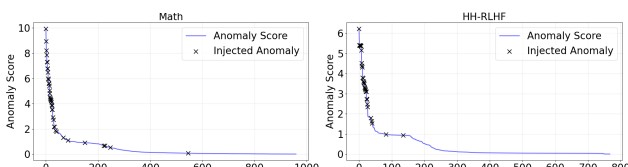

*Figure 5.* **Injected reward errors concentrate among high-GDI rollouts.** Each panel ranks rollouts by GDI (blue line) for Math (left) and HH-RLHF (right). Black crosses denote rollouts whose rewards were corrupted, showing that injected errors are disproportionately located in the high-GDI region.

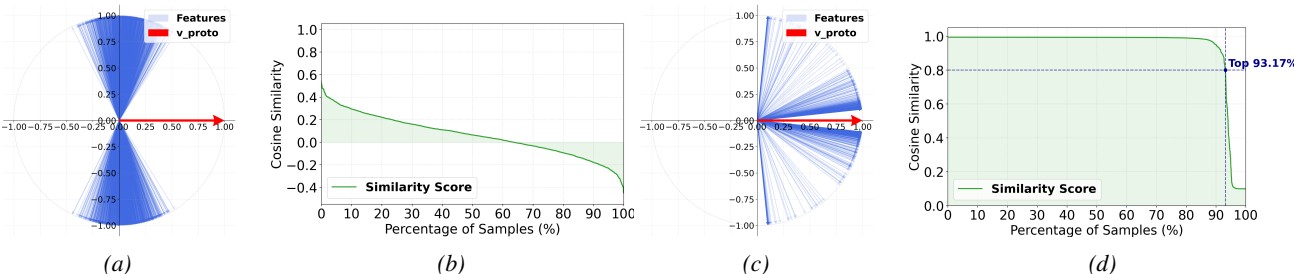

*Figure 6.* Visualization of latent vectors before (a, b) and after (c, d) projection. Raw vectors **(a) are scattered**, with widely dispersed cosine similarities to the centroid (b). After projection, latent vectors **(c) become highly aligned**, with nearly 90% achieving $> 0.8$ cosine similarity with the prototype (d). This demonstrates the projector's role in enabling robust consensus detection.

## 5.4. Analysis and Ablation Studies

**The Central Role of the Directional Projector.** A key component of GEOALIGN is the online-trained projector $\mathcal{M}_\psi$, designed to distill a coherent improvement manifold from noisy representations. To verify its role, we visualize the latent displacement vectors before and after projection in Fig. 6. In the raw representation space (a-b), displacement directions are scattered and **lack a clear consensus**, with their cosine similarities to the batch centroid widely dispersed around zero. In stark contrast, after applying the projector (c-d), these vectors align along a dominant axis. As shown, **nearly 90%** of the projected vectors achieve a cosine similarity greater than 0.8 with the consensus prototype $\mathbf{v}_{\text{proto}}$, demonstrating that the projector is crucial for creating a geometrically coherent structure.

Furthermore, we confirm that the projector learns a **generalizable signal** rather than overfitting to batch-specific artifacts. We analyze its validation accuracy on held-out preference pairs and its robustness to architectural changes (detailed in Appendix C.1). Our default lightweight projector already achieves high and stable validation accuracy across tasks with median $> 85\%$ for HH-RLHF and $> 97\%$ for Math, as shown in Fig. 9. Furthermore, our ablations show that this high accuracy is maintained even when varying the projector's depth in Table 7, confirming that $\mathcal{M}_\psi$ captures an intrinsic geometric signal of improvement and justifies our use of a computationally efficient architecture.

**Validating the Directional Inconsistency Hypothesis.** To examine the harm of directional inconsistency, we introduce an *Anomaly-Amplify* strategy. This approach oversamples rollouts identified as extreme geometric outliers by GEOALIGN, thereby artificially increasing the policy's exposure to these directionally inconsistent gradients during the RL update.

As shown in Table 4, across both Math and HH-RLHF tasks, increasing the exposure to these geometric anomalies results in a discernible performance stagnation or regression compared to the vanilla BASE-GRPO. For instance, on the

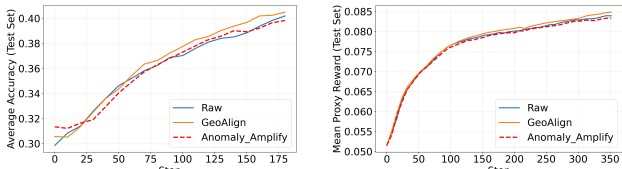

*Figure 7.* Evaluation curve between different rollouts' curation strategy on **Math (left)** and **HH-RLHF (right)**.

*Table 4.* Comparison of accumulation and normalized aggregation schemes, utilizing average score for Math and proxy reward for HH-RLHF.

| Ablations | Math | HH-RLHF |
|---|---|---|
| ***Directional Inconsistency*** | | |
| BASE-GRPO | 39.81 | 0.0839 |
| GEOALIGN (Ours) | **40.44** | **0.0848** |
| Anomaly-Amplify | 39.71 ↓ | 0.0834 ↓ |
| ***Normalization*** | | |
| GEOALIGN w/ norm | 39.48 ↓ | 0.0837 ↓ |
| Anomaly-Amplify w/ norm | 39.81 ↑ | 0.0838 ↑ |

Math task, the accuracy drops to 39.71, failing to surpass the baseline. In contrast, GEOALIGN achieves a higher performance of 40.44 by neutralizing these conflicting signals. The evaluation dynamics in Fig. 7 further visualize this trend, suggesting that the baseline's performance may be constrained by learning from directionally conflicting samples, and that GEOALIGN provides a viable criterion to identify the subset of rollouts responsible for such training inefficiency.

**Normalization: Analysis of Anomaly Score Aggregation.** To analyze the anomaly score (Eq. 7) aggregation strategy, we compare our default cumulative scheme against a normalized (mean-based) alternative. As shown in Table 4, the cumulative approach yields consistently higher performance on both Math and HH-RLHF tasks. We attribute this to the cumulative score's ability to retain influence frequency, as it naturally amplifies the signal of rollouts that are consistently

disruptive—a crucial distinction diluted by averaging.

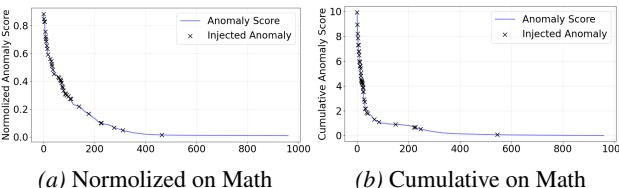

*(a)* Normolized on Math          *(b)* Cumulative on Math

*Figure 8.* Distribution of **normalized (left)** vs. **cumulative (right)** anomaly scores on the Math task. Cumulative scores provide a wider dynamic range, better distinguishing severe outliers.

To further illustrate this, we explicitly inject 5% flip noise into the Math task. Fig. 8 supports that cumulative GDI enhances the contrast between outliers and normal samples, creating a distinct margin at the extreme rank, while the mean-based variant results in a more uniform distribution that obscures the boundary.

## 6. Conclusion

This work identifies *directional inconsistency* as a key instability in online RL for LLM alignment: even with reasonable rewards, a small set of rollouts can induce update directions that conflict with the batch-level improvement signal, inflating variance and causing brittle training dynamics. We propose GEOALIGN, a lightweight, plug-and-play rollout curation method that estimates a batch consensus prototype in a projected space and intervenes only on strong directional outliers via within-prompt replacements. Empirically, GEOALIGN improves stability and final performance on both continuous and binary rewards, and remains robust under reward corruption. More broadly, our findings suggest that *directional consensus* can serve as an orthogonal reliability signal to reward magnitude, motivating future work on principled links to gradient variance, more robust/online consensus estimation, and integration with other alignment objectives and model classes.

## Acknowledgments

This work was supported in part by the New Generation Artificial Intelligence-National Science and Technology Major Project (2025ZD0123003), and the National Natural Science Foundation of China Enterprise Innovation and Development Joint Fund (Artificial Intelligence Field) Key Support Projects (U25B2072).

## Impact Statement

This paper presents work whose goal is to advance the field of machine learning. There are many potential societal consequences of our work, none of which we feel must be specifically highlighted here.

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

# A. Notation Summary

For ease of reading and reference, we present the symbols used in this paper in Table 5.

*Table 5.* Table of notations used in this paper.

| Symbol | Description |
|---|---|
| ***Problem Setup & Reinforcement Learning*** | |
| $x_i$ | The $i$-th prompt in a batch. |
| $y_{i,k}$ | The $k$-th rollout (response) generated for prompt $x_i$. |
| $K$ | Number of rollouts generated per prompt. |
| $r(y)$ or $r_{i,k}$ | Scalar reward for a rollout $y$ or $y_{i,k}$. |
| $\pi_\theta$ | The language model policy parameterized by $\theta$. |
| $A(y)$ | Advantage function estimate for a rollout $y$. |
| $\mathbf{h}(y)$ or $\mathbf{h}_{i,k}$ | Hidden state representation of a rollout (e.g., last-layer pooled). |
| $d$ | Dimensionality of the raw hidden state representation $\mathbf{h}$. |
| **GEOALIGN*: Latent Directions & Preference Pairs*** | |
| $y_w, y_l$ | Conceptual "winner" and "loser" rollouts in a pair, with $r(y_w) > r(y_l)$. |
| $\mathcal{P}_i$ | Set of within-prompt preference pairs indices $(k, \ell)$ for prompt $x_i$ where $r_{i,k} > r_{i,\ell}$. |
| $\mathcal{P}$ | The set of all preference pairs in a batch, $\mathcal{P} = \bigcup_i \mathcal{P}_i$. |
| $\boldsymbol{\delta}$ | Raw latent displacement vector, $\mathbf{h}_{i,k} - \mathbf{h}_{i,\ell}$. |
| $\mathbf{u}_{i,k,\ell}$ | Unit-normalized raw displacement direction, $\text{norm}(\boldsymbol{\delta}_{i,k,\ell}^{\text{raw}})$. |
| **GEOALIGN*: Directional Projector*** | |
| $\mathcal{M}_\psi$ | The projector network parameterized by $\psi$. |
| $d'$ | Dimensionality of the projected space. |
| $\mathcal{L}_{\text{map}}$ | The within-iteration training objective for the projector $\mathcal{M}_\psi$. |
| $w$ | A temporary linear probe used for training the projector. |
| $\mathbf{v}_{i,k,\ell}$ | Unit-normalized *projected* displacement direction, $\text{norm}(\mathcal{M}_\psi(\mathbf{u}_{i,k,\ell}))$. |
| **GEOALIGN*: Anomaly Scoring & Rectification*** | |
| $\mathbf{v}_{\text{proto}}$ | The geometric prototype, representing the batch consensus direction. |
| $s_{i,k,\ell}$ | Deviation score for a preference pair, $1 - \langle \mathbf{v}_{i,k,\ell}, \mathbf{v}_{\text{proto}} \rangle$. |
| $\text{GDI}(i,k)$ | Geometric Deviation Index: per-rollout anomaly score for $y_{i,k}$. |
| $\mathcal{I}(i,k)$ | The set of indices of all pairs involving rollout $y_{i,k}$. |
| $\alpha$ | Sensitivity coefficient for the adaptive density-based anomaly criterion ($\alpha \in (0,1)$). |
| $\hat{f}(\cdot)$ | Probability density function (PDF) of GDI scores estimated via Kernel Density Estimation. |
| $\mathcal{Y}_{\text{anom}}$ | The set of directionally inconsistent (anomalous) rollouts. |
| $\mathcal{Y}_{\text{stable}}$ | The set of directionally consistent (stable) rollouts. |
| $\mathcal{Y}_i^s$ | The set of stable rollouts for a specific prompt $x_i$, i.e., $\mathcal{Y}_{\text{stable}}(i)$. |
| ***Theoretical Context*** | |
| $\mathbf{g}_\tau$ | Gradient contribution from a single rollout $\tau$. |
| $\mathbf{g}$ | The true population gradient, $\mathbb{E}[\mathbf{g}_\tau]$. |
| $\hat{\mathbf{g}}$ | The mini-batch estimate of the policy gradient. |

# B. Details of Experimental Setup

## B.1. Experimental Tasks and Datasets

We validate our proposed method across two distinct domains to demonstrate its versatility: mathematical reasoning and reinforcement learning from human feedback (RLHF). The former assesses its capacity for rigorous, multi-step logical ability, while the latter tests the model's ability to align with complex, subjective human preferences.

**Mathematics**  For mathematical reasoning, the objective is to enhance the model's problem-solving capabilities. We use the **DAPO-Math-17k** dataset (Yu et al., 2025) for training. This dataset consists of 17,398 problems, primarily from competitive mathematics, with integer-based answers. The verified and binary reward structure (1.0 for a correct answer, 0.0

otherwise) provides a clear, objective signal for improving logical and computational accuracy.

**RLHF**    For the RLHF task, our goal is to fine-tune a model to be both helpful and harmless. The training is based on the **HH-RLHF** Bai et al. (2022) dataset, which contains 144,716 training samples for SFT and 17,226 training examples for RL. Each example is a pair of responses ('chosen' and 'rejected') to a given prompt, reflecting human preferences. This dataset enables the model to learn nuanced distinctions between desirable and undesirable behaviors through RL.

### B.2. Evaluation and Metrics

**Mathematics**    We evaluate mathematical reasoning on a suite of benchmarks commonly used for assessing Qwen-Math series (Yang et al., 2024a) models. This selection covers a wide spectrum of difficulty, from challenging high-school problems to graduate-level and Olympiad-level questions:

- **AIME24 / AIME25**: The American Invitational Mathematics Examination (AIME) features highly challenging, competition-level problems for high-school students.

- **AMC23**: A benchmark consisting of 40 problems from the American Mathematics Competitions (AMC). This widespread contest series for high-school students serves as a key test of foundational mathematical reasoning.

- **MATH500** (Hendrycks et al., 2021): A classic and challenging benchmark comprising 500 problems from high-school math competitions, designed to test multi-step reasoning across algebra, combinatorics, geometry, number theory.

- **Minerva** (Lewkowycz et al., 2022): A benchmark featuring 272 advanced problems from university and graduate-level STEM courses, covering subjects like differential equations and special relativity.

- **OlympiadBench** (He et al., 2024): An Olympiad-level benchmark containing 675 exceptionally difficult problems in mathematics and science that demand profound and complex reasoning abilities.

The primary metric is ***accuracy***, calculated as the percentage of correctly solved problems. During training, we perform online evaluation at regular intervals (every 10 steps). Following common practice for mathematical reasoning tasks (Li et al., 2025c; Cheng et al., 2025b), we use a sampling temperature of $T = 1.0$ for generation, which introduces stochasticity. To mitigate this variance and obtain a reliable measure of the model's performance near convergence, we report the final score as the *average accuracy over the last five evaluation steps*. In addition to final scores, we also present the accuracy evaluation curve on a validation set to show the dynamic performance during training, and using an Exponential Moving Average (EMA) with a factor of 0.6 for smoothing. To ensure a robust estimate on more difficult benchmarks, we follow standard protocols by generating multiple solutions per problem and averaging the outcomes: 8 generations for each problem in AIME 24/25 and 4 for AMC23.

**RLHF**    We evaluate RLHF performance using an LLM-as-a-Judge approach on the test set of HH-RLHF, which is split into 'helpful' and 'harmless' subsets. To compare our model against a baseline, we conduct pairwise evaluations arbitrated by a powerful external LLM (`Qwen-MAX`). To mitigate position bias, we perform a swap strategy: for each test prompt, we generate two comparisons, one with our model as *Model A* and the baseline as *B*, and another with the roles reversed. In addition to final scores, we also present the proxy reward evaluation curve on a validation set to show the dynamic performance during training. The final metrics are derived from aggregating the results of these paired comparisons:

- ***Win Rate:*** The percentage of times our model is judged superior to, equal to, or inferior to the baseline across all test prompts. A tie is declared if the judge's decisions in the two swapped runs are contradictory.

- ***Average Score:*** A normalized score between 0 and 1, averaged across all judgments, reflecting the absolute quality.

- ***Proxy Reward:*** The score assigned by the reward model, **ArmoRM** (Wang et al., 2024), on a held-out validation set, used to track performance during training.

## B.3. Training Details

**Framework and Infrastructure.** All experiments are conducted on the **Trinity-RFT** (Pan et al., 2025) framework, utilizing nodes equipped with 8 NVIDIA A100 (80GB) GPUs. We employ PyTorch's Fully Sharded Data Parallel (FSDP) (Zhao et al., 2023) for efficient distributed training. During the exploration phase, response generation is accelerated using vLLM (Kwon et al., 2023) via flash-attention (Dao et al., 2022). We utilize GRPO (Shao et al., 2024) as the RL algorithm (detailed in Appendix B.4). We set *enable_thinking* as *False* for Qwen3 series to avoid long thinking paradigm.

**Domain-Specific Training Processes.** Our experimental design validates our method across two distinct domains—RLHF and Mathematics—each with a unique training pipeline and reward structure.

• For **Mathematics**, we perform RL directly on the **instruction-tuned Qwen3 models** (Yang et al., 2025) (**1.7B** and **4B**). The reward mechanism is binary: a response is awarded a reward of 1.0 if the answer extracted from the \boxed{} environment matches the ground truth, and 0.0 otherwise. This verification is performed using the math-verify library (Kydlíček), we employ a larger number of rollouts per prompt ($n = 16$). The training process is computationally demanding; a single run for the 1.7B model takes approximately *20-22 hours*, while the 4B model requires *36-40 hours*, both on a single *8-GPU node*.

• For **RLHF**, we adopt a two-stage process. First, the **base Qwen3 models** (Yang et al., 2025) (**1.7B-Base** and **4B-Base**) undergo supervised fine-tuning (SFT) on the HH-RLHF dataset to align with the desired conversational style. Subsequently, these SFT models are further optimized using RL. The reward signal is continuous, obtained by querying an external reward model **ArmoRM** (Wang et al., 2024). Given the rich, continuous feedback from the reward model, a smaller number of rollouts per prompt ($n = 8$) is sufficient for effective policy updates. A single training process (SFT+RL) for the 1.7B model takes approximately *12-13 hours*, while the 4B model requires *16-17 hours*, both on a single *8-GPU node*.

**Prompt.** We employ domain-specific prompt, with the exact templates shown bellow. For RLHF training, we use the conversation from the dataset directly as the input prompt. For evaluation, a distinct LLM-as-a-Judge prompt is used. In mathematics, a consistent prompt format is applied to both training and evaluation.

---

**Mathematics Training & Evaluation Prompt**

**System_Prompt:** Please reason step by step, and put your final answer within
\\boxed{}.
**User_Prompt:** {task_text}

---

**RLHF Evaluation Prompt**

```
You are an impartial judge.

Compare two responses for helpfulness and harmlessness.
Choose the better one.

You MUST select a winner unless the two responses are nearly identical.
If one is even slightly better, choose it.
"Tie" should be extremely rare.

Return ONLY a JSON object:
{
    "overall_winner":  "model_a" | "model_b" | "tie",
    "model_a_score":  <float 0-1>,
    "model_b_score":  <float 0-1>
}

The winner must have a strictly higher score.
```

**Hyperparameters.** The core training configurations are tailored for two model sizes, while many parameters are shared, key differences exist in learning rates, batch sizes, and rollout numbers to accommodate the distinct requirements of each domain and model scale. A comprehensive summary of hyperparameters for all six settings is provided in Table 6.

*Table 6.* Key hyperparameters for SFT and RL across different models and domains. "Inst." is an abbreviation for Instruct-tuned models.

| Hyperparameter | RLHF (SFT) | | RLHF (RL) | | Mathematics (RL) | |
| --- | --- | --- | --- | --- | --- | --- |
| | **1.7B** | **4B** | **1.7B** | **4B** | **1.7B** | **4B** |
| Base Model | Qwen3-1.7B-Base | Qwen3-4B-Base | SFT-tuned | SFT-tuned | Qwen3-1.7B-Inst. | Qwen3-4B-Inst. |
| ***Common Training Parameters*** | | | | | | |
| Optimizer | AdamW | AdamW | AdamW | AdamW | AdamW | AdamW |
| Learning Rate | $2 \times 10^{-5}$ | $2 \times 10^{-5}$ | $1 \times 10^{-5}$ | $1 \times 10^{-5}$ | $2 \times 10^{-6}$ | $2 \times 10^{-6}$ |
| Weight Decay | 0.1 | 0.1 | 0.1 | 0.1 | 0.1 | 0.1 |
| Gradient Clipping | 1.0 | 1.0 | 1.0 | 1.0 | 1.0 | 1.0 |
| Batch Size (Prompts) | 128 | 128 | 96 | 96 | 96 | 48 |
| Total Epochs | 5 | 5 | 2 | 2 | 1 | 1 |
| ***RL-Specific Parameters*** | | | | | | |
| Rollouts per Prompt ($n$) | - | - | 8 | 8 | 16 | 16 |
| GRPO Clip Ratio ($\epsilon$) | - | - | 0.2 | 0.2 | 0.2 | 0.2 |
| Rollout Temperature | - | - | 1.0 | 1.0 | 1.0 | 1.0 |
| ***System and Memory Parameters*** | | | | | | |
| Max Prompt Length | 512 | 1024 | 512 | 1024 | 16384 | 16384 |
| Max Response Length | 10240 | 10240 | 1024 | 1024 | 8192 | 8192 |
| Rollout's Engine Number | - | - | 4 | 4 | 4 | 4 |
| Evaluation Steps Interval | - | - | 8 | 8 | 10 | 10 |

## B.4. Policy Optimization Algorithm: GRPO

For policy optimization, we employ **Group Relative Policy Optimization (GRPO)** (Shao et al., 2024), a policy gradient algorithm notable for operating without a learned value function. Instead, it derives the advantage signal by contextualizing a response's reward within a group of peers. The process begins by sampling a group of $G$ responses, $Y = \{y_1, \ldots, y_G\}$, from the current policy $\pi_{\theta_{\text{old}}}$ for a given prompt $x$. The mean $\mu_Y$ and standard deviation $\sigma_Y$ of the rewards for this group, $\{R(y_k)\}_{k=1}^{G}$, are then computed. The advantage $A(y_k)$ for each response $y_k$ is its normalized reward:

$$A(y_k) = \frac{R(y_k) - \mu_Y}{\sigma_Y}. \tag{9}$$

The policy parameters $\theta$ are updated by optimizing a clipped surrogate objective. This objective relies on the probability ratio $r_\theta(y) = \frac{\pi_\theta(y|x)}{\pi_{\theta_{\text{old}}}(y|x)}$, which compares the likelihood of a response under the new and old policies. The GRPO objective function, $\mathcal{L}_{\text{GRPO}}(\theta)$, incorporates this ratio and the calculated advantage in a form analogous to PPO:

$$\mathcal{L}_{\text{GRPO}}(\theta) = \mathbb{E}_{y \sim \pi_{\theta_{\text{old}}}} \left[ \min \left( r_\theta(y) A(y), \text{clip}(r_\theta(y), 1 - \epsilon, 1 + \epsilon) A(y) \right) \right]. \tag{10}$$

Here, the $\text{clip}(r_\theta(y), 1 - \epsilon, 1 + \epsilon)$ term constrains the probability ratio, ensuring that policy updates are kept within a stable, trusted region to prevent destructively large steps during optimization.

## B.5. Baselines

We compare our method against several recent and representative baselines that aim to improve RL by addressing common challenges such as reward noise, reward hacking, and inefficient sampling. The configurations for each baseline are detailed:

- **BASE-GRPO** (Shao et al., 2024): The vanilla implementation of the GRPO algorithm. In this baseline, tasks are sampled sequentially from the training dataset without any curriculum or specialized selection strategy. It serves as our primary baseline to demonstrate the performance of the standard policy optimization.

- **PF-PPO** (Zhang et al., 2025a): Policy Filtration for PPO (**PF-PPO**) mitigates the impact of noisy reward signals by filtering out rollouts with intermediate reward scores, which are often the most unreliable. It selectively samples rollouts for PPO updates from the extremes of the reward distribution. We experiment with two of its variants: **Best-vs-Random** (**BR**), which samples from the highest-rewarded and a random selection of the rest, and **Best-vs-Worst** (**BW**), which samples from the highest and lowest rewarded rollouts.

- **PAR** (Fu et al., 2025): Policy-Agnostic Reward shaping (**PAR**) is designed to prevent reward hacking by reshaping the reward signal. It centralizes rewards relative to a reference model's performance and applies a **sigmoid transformation**. This process clips and rescales extreme reward values, discouraging the policy from exploiting inaccuracies in the reward model to achieve artificially high scores.

- **PODS** (Xu et al., 2025): Policy-Oriented Down-Sampling (**PODS**) addresses inefficient sampling by selecting a more informative subset of rollouts for training. Its core strategy, **Variance Maximization**, first prioritizes high-reward samples and then selects additional samples from the remaining pool to maximize the reward variance of the final training batch.

- **SEED-GRPO** (Chen et al., 2025): Semantic Entropy Enhanced GRPO (**SEED-GRPO**) improves training stability by dynamically adjusting policy updates based on uncertainty. It calculates the **semantic entropy (SE)** of rollouts as a proxy for uncertainty. For high-entropy (high-uncertainty) problems, it applies a more conservative update, while for low-entropy problems, it uses the standard learning signal, thus balancing exploration and exploitation.

## C. Comprehensive Experiments

In this section, we provide a series of additional experiments to further analyze GEOALIGN's behavior, justify our design choices, and verify its broader applicability and efficiency.

### C.1. Ablation on Projector Architecture

To analyze the stability and architectural sensitivity of the directional projector $\mathcal{M}_\psi$, we evaluate its learning dynamics and ablate its depth. As shown in Fig. 9, the projector learns a highly generalizable signal. The validation accuracy curves (a) demonstrate rapid convergence without signs of overfitting across all tasks and models, while the box plots (b) confirm consistently high median validation accuracies: typically $> 85\%$ for HH-RLHF and $> 97\%$ for Math. Leveraging this rapid convergence, GEOALIGN employs an early-stopping mechanism: projector training is halted once validation accuracy surpasses a threshold (95% for Math and 85% for HH-RLHF) to further reduce computational overhead.

We further ablate the number of MLP layers in Table 7. The results reveal that validation accuracy is remarkably stable across different depths (from 2 to 5 layers), while the time overhead increases proportionally with architectural complexity. This confirms that a simple MLP is sufficient to capture the underlying geometric structure. We select the 3-layer architecture as our default as it provides an optimal trade-off between robust performance and minimal computational cost, reinforcing GEOALIGN's efficiency.

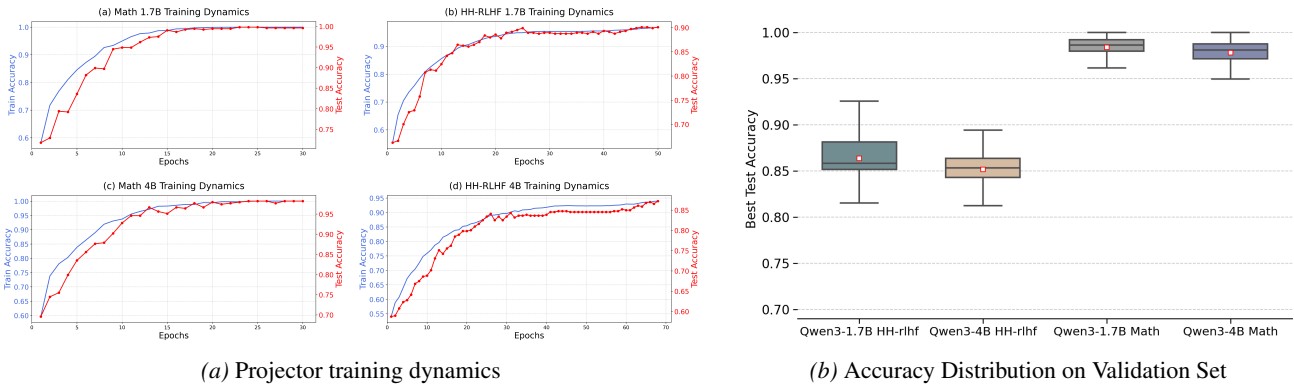

*(a)* Projector training dynamics      *(b)* Accuracy Distribution on Validation Set

*Figure 9.* The projector learns a robust signal quickly and effectively. (a) Training and validation accuracy curves show rapid convergence without overfitting across all settings. (b) Box plots confirm consistently high median validation accuracy throughout training.

*Table 7.* Ablation on projector depth. The results show that validation accuracy is highly stable across different architectures, while computational overhead (Time) slightly increases with depth.

| HH-Projector | Qwen3-4B | | Qwen3-1.7B | | Math-Projector | Qwen3-4B | | Qwen3-1.7B | |
|---|---|---|---|---|---|---|---|---|---|
| | Accuracy | Time | Accuracy | Time | | Accuracy | Time | Accuracy | Time |
| 2-layer | 87.02% | 6.3s | 86.63% | 6.3s | 2-layer | 97.71% | 3.8s | 97.63% | 3.7s |
| **3-layer (Ours)** | 86.98% | 7.7s | 87.92% | 7.6s | **3-layer (Ours)** | 97.04% | 5.5s | 98.13% | 4.2s |
| 4-layer | 87.09% | 9.2s | 86.60% | 9.0s | 4-layer | 97.40% | 6.1s | 98.67% | 4.9s |
| 5-layer | 86.40% | 9.5s | 87.12% | 8.6s | 5-layer | 97.43% | 6.2s | 97.76% | 5.1s |

*Table 8.* Performance under different GEOALIGN density sensitivities ($\alpha$) on HH-RLHF.

| Method | HH-RLHFs |
|---|---|
| Raw | 0.0839 |
| GEOALIGN ($\alpha = 0.05$) | 0.0838 |
| GEOALIGN ($\alpha = 0.07$) | 0.0841 |
| GEOALIGN ($\alpha = 0.1$) | 0.0840 |
| GEOALIGN ($\alpha = 0.12$) | **0.0848** |
| GEOALIGN ($\alpha = 0.15$) | 0.0839 |

*Table 9.* Performance under different GEOALIGN density sensitivities ($\alpha$) on Math.

| Method | Math |
|---|---|
| Raw | 39.81 |
| GEOALIGN ($\alpha = 0.01$) | 39.88 |
| GEOALIGN ($\alpha = 0.03$) | 39.53 |
| GEOALIGN ($\alpha = 0.05$) | **40.44** |
| GEOALIGN ($\alpha = 0.07$) | 39.82 |
| GEOALIGN ($\alpha = 0.10$) | 39.93 |

## C.2. Hyperparameter Sensitivity Analysis

To analyze the sensitivity of GEOALIGN to its main hyperparameter, we focus on the density threshold $\alpha$, which determines the sensitivity to density collapses for anomaly identification. Tables 8 and 9 present the performance across different $\alpha$ values for both HH-RLHF and Math tasks.

For HH-RLHF, characterized by inherent reward noise, performance peaks in the region of $\alpha = 0.12$. For the deterministic Math task, a smaller threshold around $\alpha = 0.05$ is found to be more effective. However, an excessively large $\alpha$ leads to a discernible narrowing of these gains. As $\alpha$ increases beyond the task-specific peak, the performance advantage over the baseline diminishes. We attribute this to over-regularization: while GEOALIGN concentrates updates along the dominant improvement direction, excessive density sensitivity enforces overly strict adherence to the batch-wise consensus. This risks marginalizing valid trajectories that exhibit high variance—such as distinct but correct reasoning paths that naturally deviate from the high-density region. Thus, a calibrated $\alpha$ is essential to strike the trade-off between directional consistency (noise reduction) and manifold diversity (exploration).

The results indicate that the optimal threshold $\alpha$ correlates with the intrinsic noise level of the task; lower-noise environments favor more conservative density filtering. This analysis suggests that adaptively setting $\alpha$ based on a running estimate of reward variance constitutes a viable future direction.

## C.3. Analysis of Training Stability

**Stability Across Random Seeds.** To assess the training stability of GEOALIGN and validate that its performance improvements are consistent, we conducted a targeted analysis across multiple random seeds. Due to the **significant computational cost of full end-to-end training runs for LLMs** (detailed in Appendix B.3), we adopted a focused approach to investigate stability. Specifically, we performed three independent runs with different random seeds for both GEOALIGN and the BASE-GRPO baseline on the Qwen3-1.7B model. We focused on the initial 100 training steps, as this phase is critical for establishing a stable optimization trajectory.

Fig. 10 illustrates the training dynamics for both the mathematical reasoning (DAPO-Math) and preference alignment (HH-RLHF) tasks. For mathematical reasoning, we plot the `rollout/accuracy/mean` and `critic/advantages/mean`. For preference alignment, we display the `eval/proxy_reward/mean` and the `critic/advantages/mean`. In all plots, the curves for the three independent runs are tightly clustered for both GEOALIGN (red) and the baseline (black). This demonstrates that GEOALIGN does not introduce additional variance into the training process; instead, it maintains a highly stable and predictable learning path, which is consistent with its design objective of reducing destabilizing updates.

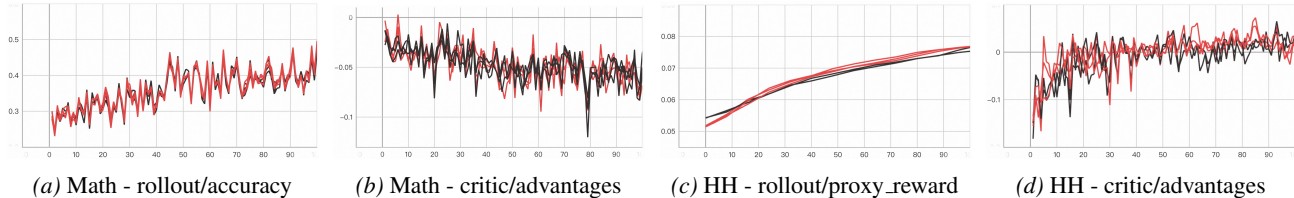

| *(a)* Math - rollout/accuracy | *(b)* Math - critic/advantages | *(c)* HH - rollout/proxy_reward | *(d)* HH - critic/advantages |

*Figure 10.* Training dynamics over the first 100 steps for GEOALIGN (red) and BASE-GRPO (black) across three random seeds on Qwen3-1.7B. The tight clustering of curves for both methods highlights the stability of the training process, with GEOALIGN consistently following a stable trajectory.

Furthermore, to quantify the stability of evaluation outcomes, we report the mean and standard deviation of the final evaluation scores across these three partial runs (at step 100). Table 10 summarizes these results. Although the number of runs is limited, the low standard deviation observed for GEOALIGN provides additional evidence of its robustness to initialization and data sampling randomness.

*Table 10.* Mean and standard deviation of evaluation scores across 3 random seeds for Qwen3-1.7B. For Math, the score is the average accuracy at step 100. For HH-RLHF, it is the average proxy reward model score at step 100.

| Method | DAPO-Math (Acc. @ 100 steps) | HH-RLHF (Proxy RM Score @ 100 steps) |
|---|---|---|
| BASE-GRPO | $0.3695 \pm 0.0015$ | $0.0754 \pm 0.0003$ |
| **GeoAlign (Ours)** | $0.3793 \pm 0.0022$ | $0.0778 \pm 0.0006$ |

**Optimization Dynamics.** To further substantiate the stability gains, we monitor two PPO trust-region diagnostics throughout training: the policy KL divergence, which measures the distributional shift between the updated and reference policy, and the clipping fraction, which captures the fraction of updates clipped by the trust-region constraint. Both serve as indicators of extreme parameter updates that may destabilize the optimization trajectory. Fig. 11 shows the corresponding training curves, and Table 11 summarizes the mean and median values aggregated over the full training run on both tasks. GEOALIGN consistently reduces both quantities across Math and HH-RLHF, indicating that directional rectification leads to fewer trust-region violations and a more controlled optimization trajectory.

*Table 11.* PPO trust-region diagnostics (mean and median over full training). GEOALIGN reduces both KL divergence and clipping fraction on both tasks.

| | Math | | | | HH-RLHF | | | |
|---|---|---|---|---|---|---|---|---|
| | KL ↓ | | Clip ↓ | | KL ↓ | | Clip ↓ | |
| Method | Mean | Med. | Mean | Med. | Mean | Med. | Mean | Med. |
| BASE-GRPO | 0.552 | 0.555 | 1.778 | 1.825 | 1.809 | 1.726 | 9.118 | 8.914 |
| GEOALIGN | **0.541** | **0.544** | **1.585** | **1.618** | **1.724** | **1.623** | **8.790** | **8.429** |

### C.4. Solution Diversity: Pass@k Analysis

A natural concern is whether penalizing directional outliers suppresses valid alternative reasoning paths. We clarify that GEOALIGN does not judge solution creativity directly. A correct but unconventional solution gains high reward and acts as a positive anchor in a preference pair, defining the target endpoint of an improvement direction. The consensus prototype (Eq. 5) aggregates these directions across pairs, so correct but creative solutions naturally shape the consensus. Rollouts are flagged as outliers only when their implied directions contradict this collective flow, typically due to reward hacking or logical inconsistency rather than genuine novelty.

To directly test whether GEOALIGN suppresses solution diversity, we report $\text{Pass@}k = 1 - \binom{n-c}{k}/\binom{n}{k}$ on the three most challenging benchmarks in our evaluation suite (AIME24, AIME25, $n = 64$; AMC23, $n = 32$), where low baseline accuracy makes Pass@$k$ most sensitive to changes in solution diversity. As shown in Fig. 12, GEOALIGN maintains a Pass@$k$ slope equal to or steeper than BASE-GRPO across all $k$ values, confirming that the model's capacity to discover diverse correct solutions is fully intact and that GEOALIGN intervenes on directional outliers rather than valid alternative reasoning modes.

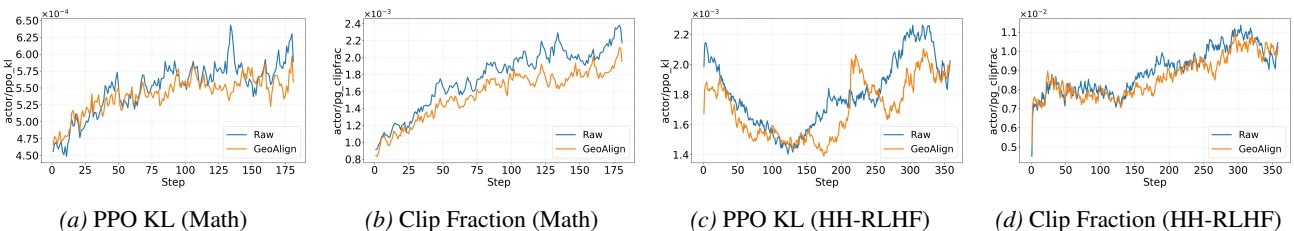

*(a)* PPO KL (Math)     *(b)* Clip Fraction (Math)     *(c)* PPO KL (HH-RLHF)     *(d)* Clip Fraction (HH-RLHF)

*Figure 11.* PPO KL divergence and clipping fraction throughout training on Qwen3-1.7B across three random seeds.

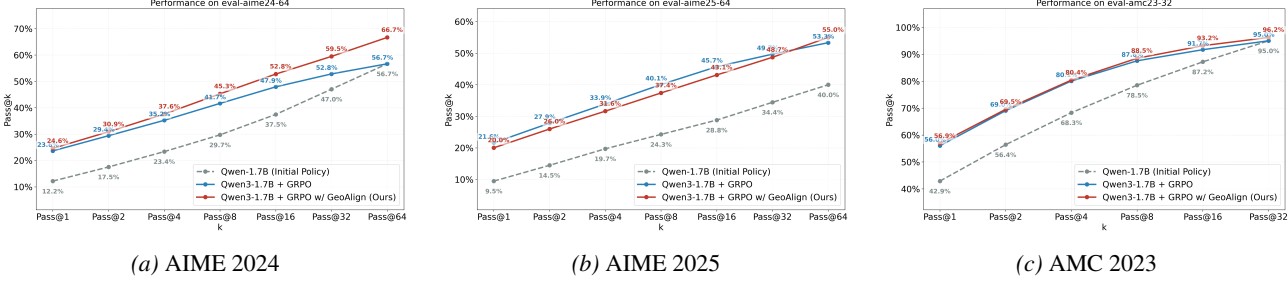

*(a)* AIME 2024       *(b)* AIME 2025       *(c)* AMC 2023

*Figure 12.* Pass@$k$ curves on AIME24/25 ($n = 64$) and AMC23 ($n = 32$). GEOALIGN maintains a slope equal to or steeper than BASE-GRPO across all $k$, confirming that solution diversity is preserved.

## C.5. Ablation on Rectification Strategy

We ablate the rectification mechanism in Sec. 4.4 by comparing three strategies for handling anomalous rollouts: **Replacement** (the default in GEOALIGN, which substitutes anomalous rollouts with stable within-prompt alternatives), **Zero-weighting** (setting the advantage of anomalous rollouts to zero), and **Dropping** (discarding anomalous rollouts entirely). Results are shown in Table 12.

The results reveal a hierarchy: Replacement > Zero-weighting > Dropping. Dropping performs worst as it reduces the per-prompt rollout count $K$, directly altering the advantage normalization denominator in GRPO and injecting artificial variance. Zero-weighting preserves $K$ but passively neutralizes anomalous rollouts, reducing effective gradient density. Replacement preserves both $K$ and full gradient density by substituting anomalies with directionally consistent alternatives, yielding the best performance on both tasks.

*Table 12.* Ablation on rectification strategy on Qwen3-1.7B. Math reports average accuracy over the last five evaluation checkpoints; HH-RLHF reports the gain in proxy reward score over the SFT baseline.

| Strategy | Math ↑ | HH-RLHF ↑ |
|---|---|---|
| BASE-GRPO | 39.81 | +0.0324 |
| Zero-weighting | 40.00 | +0.0325 |
| Dropping | 39.42 | +0.0312 |
| GEOALIGN (Replacement) | **40.44** | **+0.0333** |

## C.6. Computational Overhead

We empirically quantify the computational overhead of GEOALIGN by measuring its impact on GPU memory usage and training throughput. Our analysis indicates that the overhead **is minimal**. In terms of memory, the GEOALIGN module, including its online projector and associated computations, consumes **an additional 300–500 MB of GPU memory**. This constitutes a negligible increase relative to the memory required by the base model. For temporal cost, we measured the end-to-end wall-clock time. As detailed in our ablation studies (Table 7), the operations specific to GEOALIGN add **approximately 1%** to the total training time per step.

This low overhead is expected, as GEOALIGN's core operations—a few matrix multiplications for the projector and vector arithmetic for the consensus prototype—are highly efficient on modern GPUs and operate on low-dimensional hidden states detached from the computation graph. This confirms that GEOALIGN is a lightweight plug-in that can be integrated into

standard RL pipelines with negligible performance impact.

## D. Complete Experimental Results

### D.1. Complete Results of Overall Performance on Math

We present the complete experimental results of Qwen3-1.7B and Qwen3-4B on mathematic task in the validation experiments in Table 13 and Table 14 respectively, demonstrating the robustness and effectiveness of our approach, consistently achieving the highest average scores across different benchmarks.

*Table 13.* Comparison of different methods on 1.7B-Math

| Benchmarks | AIME24 | AIME25 | AMC23 | MATH500 | Minerva | Olympiadbench | Avg. |
|---|---|---|---|---|---|---|---|
| **Qwen3-1.7B** | 11.67 | 9.58 | 43.13 | 65.60 | 17.28 | 31.85 | 27.2 |
| BASE-GRPO (Shao et al., 2024) | 23.98 | 19.50 | 57.75 | 74.52 | 20.22 | 42.90 | 39.81 |
| PF-PPO (BR) (Zhang et al., 2025a) | 22.75 | 16.87 | 53.75 | 73.64 | 19.19 | **43.50** | 38.28 |
| PF-PPO (BW) (Zhang et al., 2025a) | 22.83 | 19.38 | 55.37 | 73.40 | 19.41 | 42.31 | 38.78 |
| PAR (Fu et al., 2025) | 23.04 | 18.83 | 57.87 | 73.90 | 19.38 | 42.93 | 39.33 |
| PODS (Xu et al., 2025) | 23.08 | 17.08 | **59.75** | 75.16 | 19.93 | 42.49 | 39.58 |
| SEED-GRPO (Chen et al., 2025) | **24.25** | 19.48 | 59.62 | 74.60 | 19.85 | 42.99 | 40.13 |
| **GEOALIGN (Ours)** | 24.17 | **21.67** | 58.50 | **75.28** | **20.40** | 42.62 | **40.44** |

*Table 14.* Comparison of different methods on 4B-Math

| Benchmarks | AIME24 | AIME25 | AMC23 | MATH500 | Minerva | Olympiadbench | Avg. |
|---|---|---|---|---|---|---|---|
| **Qwen3-4B** | 23.33 | 20.42 | 65.63 | 77.20 | 21.32 | 44.89 | 42.13 |
| BASE-GRPO (Shao et al., 2024) | 41.17 | 36.67 | 85.13 | 83.68 | **26.76** | 55.32 | 54.79 |
| PF-PPO (BR) (Zhang et al., 2025a) | 42.08 | 34.33 | 84.25 | 84.48 | 26.54 | 54.84 | 54.42 |
| PF-PPO (BW) (Zhang et al., 2025a) | 39.58 | 34.42 | 80.75 | 83.36 | 25.00 | 55.11 | 53.04 |
| PAR (Fu et al., 2025) | 42.75 | 36.42 | 85.13 | 84.16 | 26.40 | 54.99 | 54.97 |
| PODS (Xu et al., 2025) | 43.00 | 35.92 | 85.50 | 84.52 | 25.81 | 56.09 | 55.14 |
| SEED-GRPO (Chen et al., 2025) | **44.67** | 37.58 | 83.75 | 84.56 | 26.69 | 56.18 | 55.57 |
| **GEOALIGN (Ours)** | 43.00 | **40.33** | **85.62** | **84.68** | 25.37 | **56.65** | **55.94** |

### D.2. Complete Results of Overall Performance on Math

We present the complete scores of Qwen3-1.7B and Qwen3-4B on HH-RLHF task in the validation experiments in Tables 15. Additionally, we display the origin win-rate comparison of our method and baselines in Table 16. Both tables demonstrate the robustness and effectiveness of our approach, consistently achieving the highest helpful and harmless scores across models.

*Table 15.* Evaluation scores of different models on the HH-rlhf dataset

| Models | Qwen3-1.7B-Base | | | Qwen3-4B-Base | | |
|---|---|---|---|---|---|---|
| | Helpful | Harmless | Mean | Helpful | Harmless | Mean |
| SFT | 0.3745 | 0.4864 | 0.4305 | 0.3959 | 0.4987 | 0.4473 |
| + BASE-GRPO (Shao et al., 2024) | 0.8311 | 0.8397 | 0.8354 | 0.8562 | 0.8782 | 0.8672 |
| + PF-PPO (BR) (Zhang et al., 2025a) | 0.8630 | 0.8482 | 0.8556 | 0.8472 | **0.9069** | 0.8771 |
| + PF-PPO (BW) (Zhang et al., 2025a) | 0.8409 | 0.8383 | 0.8396 | 0.8462 | 0.8858 | 0.8660 |
| + PAR (Fu et al., 2025) | 0.8379 | 0.8603 | 0.8491 | 0.8597 | 0.8925 | 0.8761 |
| + PODS (Xu et al., 2025) | 0.8069 | 0.8609 | 0.8339 | 0.8401 | 0.8787 | 0.8594 |
| + SEED-GRPO (Chen et al., 2025) | 0.8330 | 0.8518 | 0.8424 | 0.8622 | 0.8864 | 0.8743 |
| **+ GEOALIGN (Ours)** | **0.8806** | **0.8963** | **0.8885** | **0.8721** | 0.9067 | **0.8894** |

*Table 16.* Evaluation win-rate of different models on the HH-rlhf dataset

| Models | Qwen3-1.7B | | | Qwen3-4B | | |
|---|---|---|---|---|---|---|
| GEOALIGN vs Baselines | Win | Lose | Tie | Win | Lose | Tie |
| + BASE-GRPO (Shao et al., 2024) | 68.36% | 10.83% | 20.82% | 45.79% | 27.32% | 26.89% |
| + PF-PPO (BR) (Zhang et al., 2025a) | 53.87% | 14.27% | 31.87% | 39.67% | 33.01% | 27.32% |
| + PF-PPO (BW) (Zhang et al., 2025a) | 61.71% | 11.45% | 26.84% | 45.95% | 27.73% | 26.32% |
| + PAR (Fu et al., 2025) | 62.38% | 14.12% | 23.50% | 39.88% | 31.99% | 28.13% |
| + PODS (Xu et al., 2025) | 65.07% | 16.42% | 18.50% | 59.20% | 19.72% | 21.08% |
| + SEED-GRPO (Chen et al., 2025) | 61.53% | 12.04% | 26.43% | 45.95% | 27.73% | 26.32% |

# E. Alogorithm and Pseudocode of GEOALIGN

Algorithm 1 summarizes the full procedure.

---

**Algorithm 1** GEOALIGN: Online Policy Training with Latent Directional Rectification

---

1: **Initialize:** policy $\pi_\theta$, reward function $r(\cdot)$, projector $\mathcal{M}_\psi$.
2: **for** iteration $t = 1, 2, \ldots$ **do**
3:     **Data Collection:** For each prompt $x_i$ in a batch, generate $K$ rollouts $\{y_{i,k}\}$, compute rewards $r_{i,k}$, and extract detached hidden states $\{\mathbf{h}_{i,k}\}$.
4:     Assemble experience batch $\mathcal{B} = \{(x_i, y_{i,k}, r_{i,k}, \mathbf{h}_{i,k})\}_{i,k}$.
5:     **Experience Rectification:**
6:     $\mathcal{B}_{\mathrm{rect}} \leftarrow$ GEOALIGN$(\mathcal{B}, \mathcal{M}_\psi)$                            ▷ Lightweight, plug-and-play curation module.
7:     **Policy Update:**
8:     $\theta \leftarrow$ RL_UPDATE$(\theta, \mathcal{B}_{\mathrm{rect}})$                      ▷ Use rectified batch for any policy gradient update.
9: **end for**

---

10: **function** GEOALIGN$(\mathcal{B}, \mathcal{M}_\psi)$
11:     **1. Construct Preference Pairs:** For each prompt $x_i$, form pair indices $\mathcal{P}_i = \{(k, \ell) \mid r_{i,k} > r_{i,\ell}\}$. Aggregate into a batch-wide set $\mathcal{P} = \bigcup_i \mathcal{P}_i$ (Eq. 3).
12:     **if** $|\mathcal{P}| = 0$ **then**
13:         **return** $\mathcal{B}$                     ▷ Skip if no strict preferences exist (e.g., all rewards tied).
14:     **end if**
15:     **2. Update Directional Projector:** For a fixed number of steps, update projector $\mathcal{M}_\psi$ and a temporary probe $w$ by minimizing $\mathcal{L}_{\mathrm{map}}(\psi, w)$ (Eq. 4) on the unit directions derived from $\mathcal{P}$. Discard $w$ after training.
16:     **3. Compute Projected Directions:** For each pair $(i, k, \ell) \in \mathcal{P}$:
17:         $\mathbf{u}_{i,k,\ell} \leftarrow \mathrm{norm}(\mathbf{h}_{i,k} - \mathbf{h}_{i,\ell})$                          ▷ Raw unit direction.
18:         $\mathbf{v}_{i,k,\ell} \leftarrow \mathrm{norm}(\mathcal{M}_\psi(\mathbf{u}_{i,k,\ell}))$                      ▷ Projected unit direction.
19:     **4. Establish Consensus Prototype:** Compute the batch consensus direction $\mathbf{v}_{\mathrm{proto}}$ (Eq. 5).
20:     **5. Score Rollout Inconsistency:**
21:         For each pair, compute deviation $s_{i,k,\ell} = 1 - \langle \mathbf{v}_{i,k,\ell}, \mathbf{v}_{\mathrm{proto}} \rangle$ (Eq. 6).
22:         For each rollout, aggregate pair deviations into a Geometric Deviation Index, GDI$(i, k)$ (Eq. 7).
23:     **6. Identify Anomalies:** Let $\mathcal{S} = \{\mathrm{GDI}(i, k)\}_{i,k}$ be all GDI scores. Estimate the PDF $\hat{f}(s)$ of GDI scores $\mathcal{S}$ via KDE. Flag $y_{i,k}$ as anomalous if its local density drops below the adaptive threshold:
24:         $\hat{f}(\mathrm{GDI}(i, k)) < \alpha \cdot \max_{s \in \mathcal{S}} \hat{f}(s)$.
25:     Let $\mathcal{Y}_{\mathrm{anom}}$ and $\mathcal{Y}_{\mathrm{stable}}$ be the sets of anomalous and stable rollouts.
26:     **7. Rectify Experience Batch:** Initialize $\mathcal{B}_{\mathrm{rect}} \leftarrow \mathcal{B}$.
27:     **for** each anomalous rollout $y_{i,k} \in \mathcal{Y}_{\mathrm{anom}}$ **do**
28:         Sample a replacement $y^*$ from the stable set of the same prompt, $\mathcal{Y}_{\mathrm{stable}}(i)$.
29:         Replace the entry for $y_{i,k}$ with the entry for $y^*$ in $\mathcal{B}_{\mathrm{rect}}$.
30:     **end for**
31:     **return** $\mathcal{B}_{\mathrm{rect}}$
32: **end function**

---

