# OpenReview forum: "GeoAlign: Geometric Rollout Curation for Robust LLM Reinforcement Learning"
_ICML.cc/2026/Conference — ICML 2026 regular_

### Official Review · Reviewer_koDT · 2026-03-11

**Soundness:** 3
**Presentation:** 3
**Significance:** 2
**Originality:** 3
**Overall Recommendation:** 4
**Confidence:** 3

**Summary:**

The paper proposes GeoAlign, a lightweight plug-in that stabilizes online reinforcement learning for LLMs by mitigating the directional inconsistency, where high-reward rollouts push the model in conflicting latent directions. Using efficient, forward-pass-only calculations, it projects these update directions, flags severe outliers that deviate from the batch consensus, and replaces them to ensure stable policy updates. It improves both training stability and resilience to noisy rewards in mathematical reasoning and RLHF tasks.

**Compliance With Llm Reviewing Policy:**

Affirmed.

**Final Justification:**

My concerns are addressed, and I have raised my score from 3 to 4.

**Key Questions For Authors:**

1. How does pass@k of raw GRPO and GeoAlign compare?

2. Is there a principled way to tune the anomaly budget $k$?

**Limitations:**

See Weaknesses.

**Strengths And Weaknesses:**

Strengths:

1. The method is efficient, working entirely on detached forward-pass hidden states and introducing little computational overhead.

2. The perspective of modeling online RL instability into a geometric directional alignment problem is intuitive.

Weaknesses:

1. The method requires additional tuning efforts on hyperparameters (e.g. the budget $k$) and projector architecture (e.g., number of layers). As Tables 7-9 in Appendix C show, the performance of GeoAlign is sensitive, and some configs underperform the raw baseline (e.g., $k=20\%$).

2. Continuing Weakness 1, the improvement seems marginal. The best improvement on math is within 1%, and some configurations result in worse performance than the baseline.

3. The GeoAlign method is based on a key assumption that the consensus direction is the "correct" direction, which is questionable. For example, for complex reasoning task, a highly creative, correct but unconventional solution may be flagged as an outlier and does not contribute to the gradient. This may result in diversity collapse. The paper does not provide entropy or pass@k metrics to examine the diversity change.

---

> ### Author Rebuttal · Authors · 2026-03-31
>
> Thank you for the constructive review and recognition of our geometric perspective and computational efficiency. We address each Weakness (W) and Question (Q) below. Tables and Figures referenced below are in our [anonymous repo](https://anonymous.4open.science/r/GeoAlign-Rebuttal-E5F7/rebuttal_experiments.md).
>
>
> ---
>
> **W1 & Q2: On Hyperparameter Sensitivity and Principled Tuning of κ**
>
> > *"performance of GeoAlign is sensitive... Is there a principled way to tune the anomaly budget κ?"*
>
> We address the concerns regarding the anomaly budget and projector architecture as follows:
>
> **$\kappa$ Sensitivity and Robustness:**
> 1. **Empirical Evidence:** The κ=20% result in Table 9 is a boundary case from our ablation study, rather than a recommended setting. As discussed in Appendix. C.2, an excessively large budget suppresses valid reasoning paths and leads to over-regularization—a predictable failure mode. Across both tasks, GEOALIGN is robust around κ=5%, consistently serving as a stable and well-supported default setting in practice.
> 2. **Projector Architecture:** Table 7 and Table R7 (anonymous repo) show that performance is *not* sensitive to projector depth. A 3-layer MLP provides sufficient representational capacity, and deeper variants yield no meaningful gain. This confirms that GEOALIGN can be deployed with fixed default configurations without additional architectural search.
>
> **Principled Tuning of $\kappa$:**
> The selection of $\kappa$ is guided by the **Conservative Intervention** principle (Sec. 3.3) and the observed long-tail geometry of **GDI scores** (Fig. 3):
> 1. **Constrained to a Small Range:** Our analysis (Sec. 4.4) reveals that directional inconsistency typically follows a heavy-tailed distribution, where severe outliers represent only a small fraction of rollouts. Consequently, $\kappa$ is naturally constrained to a small range (typically $\le$ 10%) to avoid interfering with the majority of valid updates.
> 2. **Stable Balance and Further Improvement:** Empirical results in Appendix. C.2 show that a default κ =5% provides a stable and robust balance across both clean and noisy tasks, improving training stability without sacrificing performance. While this default performs well, $\kappa$ can be further refined for additional performance gains based on reward reliability: a smaller budget ($\kappa \in$ [2%, 5%]) for high-precision rewards (e.g., Math), and a slightly larger budget ($\kappa \in$ [5%, 10%]) for noisier rewards (e.g., RLHF) to better mitigate reward artifacts.
>
> ---
>
> **W2: On the Margin of Improvement**
>
> > *"improvement seems marginal...within 1%...some configurations result in worse performance"*
>
> We respectfully offer third clarifications:
>
> 1. the sub-baseline configurations are boundary tests from our ablation (see W1), not operational settings. Under the recommended defaults, GEOALIGN consistently **outperforms both BASE-GRPO and all four baselines across all four model–task combinations** (Tables 1–2).
>
> 2. GEOALIGN's value extends beyond peak scores. Its most discriminative advantage appears under **reward noise**: in Table 3, BASE-GRPO suffers a **6.9%** relative drop under 5% Math reward corruption, while GEOALIGN degrades by only **5.1%**—the smallest among all methods.
>
> 3. GEOALIGN fundamentally **stabilizes the optimization dynamics**. We monitor pg_clipfrac (fraction of clipped updates) and KL divergence as direct indicators of optimization conflict in Tables R2-R3 & Fig. R1 (anonymous repo), and GEOALIGN consistently reduces both metrics.
>
> ---
>
> **W3 & Q1: On Consensus Direction Validity and Solution Diversity**
>
> > *"a highly creative, correct but unconventional solution may be flagged as an outlier...may result in diversity collapse... does not provide entropy or pass@k"*
>
> We clarify a key design point: GEOALIGN does not judge solution creativity directly. A correct but unconventional solution gains high-reward and acts as a **positive anchor** in a preference pair, defining the target endpoint of an improvement direction. The consensus prototype (Eq. 5) aggregates these directions across pairs, so correct but creative solutions naturally shape the consensus. Rollouts are flagged as outliers only when their implied directions contradict this collective flow—typically due to reward hacking or logical inconsistency, rather than genuine novelty.
>
> **Pass@k evidence (Fig. R2 in anonymous repo):** We report Pass@k on AIME24/AIME25 (k=64), and AMC23 (k=32)—benchmarks that inherently require diverse reasoning strategies. GEOALIGN maintains a Pass@k curve slope equal to or steeper than BASE-GRPO across all k values, confirming that valid alternative reasoning modes are preserved.
>
> ---
>
> **Closing Remarks**
>
> We thank you for the focused and principled feedback. We sincerely hope these responses address your concerns—any improvement in your evaluation score would be a great encouragement to us, and we warmly welcome any further questions.

---

> > ### Author Rebuttal · Reviewer_koDT · 2026-04-03
> >
> > Thank you for the rebuttal and the additional experimental results. My concerns have been addressed, and I will raise my score to 4 (after final justification).

---

> > > ### Author Response · Authors · 2026-04-08
> > >
> > > Thank you for your positive response and for reviewing our rebuttal and additional experimental results. We are very glad to hear that your concerns have been addressed.
> > >
> > > We also sincerely thank you for indicating that you will raise your score to 4 after final justification. As the author–reviewer discussion period is approaching its end, we would be very grateful if you could kindly complete the final justification when convenient.
> > >
> > > Thank you again for your time and support!

---

### Official Review · Reviewer_hJz8 · 2026-03-12

**Soundness:** 3
**Presentation:** 4
**Significance:** 4
**Originality:** 4
**Overall Recommendation:** 5
**Confidence:** 4

**Summary:**

The paper identifies a critical failure mode in online Reinforcement Learning for Large Language Models termed "directional inconsistency," where a small subset of high-reward rollouts induces policy update directions that sharply conflict with the batch majority in the representation space. To address this, the authors propose GEOALIGN, a lightweight, forward-pass-only rollout curation module. GEOALIGN constructs within-prompt preference pairs, learns a small online projector to map hidden states to a concentrated improvement manifold, calculates a Geometric Deviation Index based on distance to a batch consensus prototype, and replaces highly anomalous rollouts with stable alternatives.

**Compliance With Llm Reviewing Policy:**

Affirmed.

**Key Questions For Authors:**

Given that the optimal filtering ratio $\kappa$ appears correlated with the intrinsic noise level of the task and reward signal, have you explored dynamically adapting $\kappa$ and $\tau$ during training based on the running variance or distribution of the GDI scores?
Does the output dimensionality $d'$ of the projector $\mathcal{M}_\psi$ affect the concentration of the improvement manifold? Is there a risk of collapsing useful, task-relevant behavioral diversity if $d'$ is too heavily bottlenecked?

**Limitations:**

Yes.

**Strengths And Weaknesses:**

Strengths:
Conceptual Originality: Diagnosing RL instability through the geometric lens of representation space rather than merely relying on scalar reward magnitudes is a highly original and insightful perspective.
High Practical Efficiency: The design of GEOALIGN is exceptionally practical.
Robust Empirical Validation: The experimental design is rigorous. Testing the framework across completely different reward structures—deterministic binary rewards in DAPO-Math-17k and continuous, subjective reward models in HH-RLHF—proves the generalizability of the geometric signal. The controlled noise injection experiments effectively isolate and prove the mechanism's core claim.

Weaknesses:
Gap in Theoretical Justification: The paper intuitively argues that directional inconsistency correlates with the variance of stochastic policy gradients. However, it stops short of providing a formal mathematical proof linking the cosine distance in the projected latent space to the actual variance of the parameter-space gradients $\hat{g}$.
Scale of Evaluation: While testing on Qwen3-1.7B and 4B is adequate for demonstrating the algorithmic mechanics, RL training dynamics can change dramatically at larger scales (e.g., 8B, 32B, or 70B parameters). Validating whether these specific geometric properties hold uniformly at a larger scale would strengthen the paper.

---

> ### Author Rebuttal · Authors · 2026-03-31
>
> Thank you for your thorough and constructive review, and for recognizing the conceptual originality, the practical efficiency, and the rigor of our empirical validation. We address each Weakness (W) and Question (Q) below. We included Tables and Figures for rebuttal in this [anonymous repo](https://anonymous.4open.science/r/GeoAlign-Rebuttal-E5F7/rebuttal_experiments.md).
>
> ---
>
> **W1: On the Gap in Theoretical Justification**
>
> > *"stops short of providing a formal mathematical proof linking ..."*
>
> We thank the reviewer for this precise observation and acknowledge it as a current limitation. As we state explicitly in **Sec. 3.3**, the formal connection between latent directional consensus and parameter-space gradient variance remains an open theoretical question—bridging representation-space geometry to optimization dynamics is genuinely non-trivial.
>
> While a direct formal derivation remains beyond our current reach, we have sought to further substantiate the hypothesized link between latent directional deviation and parameter-space optimization conflict through additional **empirical validation**. We provide a direct optimization-level measurement: **Table R2-R3 & Fig. R1** (anonymous repo) reports the KL divergence between the updated and reference policy throughout training. GeoAlign consistently maintains a lower and more stable KL trajectory compared to BASE-GRPO.
>
> We believe a rigorous theoretical treatment is a valuable and promising future direction for this line of work.
>
> ---
>
> **W2: On the Scale of Evaluation**
>
> > *"... at larger scales ..."*
>
> We have conducted additional experiments on **Qwen3-8B** HH-RLHF, with the same hyperparameters as the main submission. Due to the limitation of resources and time, we included only one experiment comparing raw and ours, results are shown in **Table R5 & Fig. R3** (anonymous repo).
>
> Crucially, analysis of the 8B training run confirms that the **long-tail GDI distribution** persists at this scale—the phenomenon is not an artifact of smaller models. The full training curves show consistently smoother optimization compared to BASE-GRPO. We acknowledge that validation at 32B/70B would further strengthen the paper; we consider this a future direction given current resource constraints.
>
> ---
>
> **Q1: On Dynamic Adaptation of $\kappa$ and $\tau$**
>
> > *"... dynamically adapting K and T during training ..."*
>
> Our current fixed $\kappa$ and $\tau$ follow the **Budgeted Intervention** design principle (Sec. 3.3): conservative, stable intervention whose robustness is already demonstrated in **Appendix C.2** (Tables 8–9), showing that performance is stable across a reasonable range of both hyperparameters.
>
> Additionally, to explore the hyperparameters' adjustment, we implemented a **KDE-based dynamic filtering variant** as a new ablation study, which generalizes the fixed $\kappa$ and $\tau$ into a distribution-aware threshold.
>
> In this adaptive trial, we model the PDF $\hat{f}(s)$ of the current batch and flag outliers where the local density drops significantly: $\hat{f}(\text{GDI}) < \alpha \cdot \max \hat{f}(s)$. This formulation allows the effective $\kappa$ to be dynamically determined by the "slope fault" of the GDI distribution. Based on our findings in Appendix C.2, we performed a preliminary trial with $\alpha = 0.05$ for Math and $\alpha = 0.1$ for HH-RLHF. As shown in **Table R9 & Figure R5** (anonymous repo), without extensive hyperparameter tuning, this adaptive approach already outperforms Base-GRPO.
>
> ---
>
> **Q2 & Q3: On Projector Dimensionality $d'$ and Behavioral Diversity**
>
> > *"...output dimensionality $d'$ affect the concentration ... Is there a risk ... if $d'$ is too heavily bottlenecked?"*
>
> We have conducted additional analyses to address both concerns.
>
> **Empirical stability across $d'$ (Q2).** On the Math dataset, the projector maintains consistently high validation accuracy (~98%) across a four-fold range ($d' \in \{128, 256, 512\}$; **Table R8** in anonymous repo). Crucially, the GDI score distributions across different $d'$ values are virtually overlapping (**Fig. R4** in anonymous repo), and the identified outlier subsets remain functionally identical even at $d'=128$.
>
> **Diversity preservation (Q3).** To directly test whether GEOALIGN suppresses task-relevant behavioral diversity, we evaluate **Pass@k** on challenging benchmarks (AIME24, AIME25, AMC23; **Fig. R2** in anonymous repo). Under our default $d'$, GeoAlign exhibits a Pass@$k$ scaling curve that is equal to or even steeper than standard GRPO. Since outlier identification remains functionally stable across the tested range of $d'$, the risk of an overly narrow bottleneck inadvertently removing task-relevant behavioral diversity is minimal.
>
> ---
>
> **Closing Remarks**
>
> We sincerely thank you for the careful and encouraging review. We hope these responses address your remaining concerns, and we warmly welcome any further questions or discussion.

---

> > ### Author Rebuttal · Reviewer_hJz8 · 2026-04-03
> >
> > While the authors' additional 8B validation, Pass@k analysis, and optimization-level metrics substantially strengthen the empirical case, the formal theoretical link between latent directional consensus and parameter-space gradient variance remains an open question that would benefit from further discussion or framing in the final version.

---

> > > ### Author Response · Authors · 2026-04-05
> > >
> > > Thank you sincerely for your continued engagement and for the precise framing of your remaining concern. We genuinely appreciate that you have taken the time to articulate the specific gap that still needs addressing: **the formal link between latent directional consensus in the projected representation space and the variance of parameter-space policy gradients**.
> > >
> > > ---
> > >
> > > **On Our Previous Response**
> > >
> > > In our first rebuttal, we chose to provide optimization-level empirical evidence (KL divergence trajectories, Table R2–R3 & Fig. R1 in anonymous link) rather than a rigor theoretical derivation. This was a deliberate choice: we were cautious about presenting a formal argument that could not be rigorously defended, as we believed an imprecise proof would do more harm to the paper's credibility than honest acknowledgment of a theoretical frontier, and we still hold this view.
> > >
> > > ---
> > >
> > > **A Closer Look at the Theoretical Connection**
> > >
> > > We want to be transparent about where the mathematical difficulty lies, and then offer what we believe is a meaningful — if not yet complete — theoretical analysis.
> > >
> > > **The core challenge.** The parameter-space gradient contribution of a rollout $\tau$ takes the form:
> > >
> > > $$g_\tau = \hat{A}(\tau) \cdot \nabla_\theta \log \pi_\theta(\tau),$$
> > >
> > > where $\hat{A}(\tau)$ is a scalar advantage estimate and $\nabla_\theta \log \pi_\theta(\tau)$ is a high-dimensional vector that depends on the current policy parameters $\theta$ and is computed via a full backward pass. In contrast, our latent displacement vector $\delta = h(y_w) - h(y_l)$ lives in the representation space of the final hidden layer, is detached from $\theta$, and is computed in a single forward pass. Establishing a formal, general correspondence between the *direction* of $\delta$ and the *direction* of $g_\tau$ is non-trivial precisely because $\nabla_\theta \log \pi_\theta$ spans all layers of the model, while $\delta$ is local to the final hidden layer.
> > >
> > > **A tractable partial connection.** However, consider the final linear projection layer with weight matrix $W \in \mathbb{R}^{V \times d}$ (mapping hidden states to vocabulary logits). For a token at position $t$, the gradient contribution through this layer is proportional to $W^\top \cdot \Delta p_t$, where $\Delta p_t$ encodes the policy's output correction at that position. This means the *final hidden state* $h$ directly modulates the gradient signal flowing through $W$: rollouts whose hidden representations differ substantially in direction will, through $W$, induce gradient contributions that differ in direction in the parameter space of the output layer. Specifically, if two rollouts $\tau_1$ and $\tau_2$ satisfy $\langle h(\tau_1), h(\tau_2) \rangle \approx -1$ (i.e., their representations are approximately antipodal), the induced output-layer gradient contributions will also tend to oppose each other, inflating the variance of the mini-batch gradient estimator. Our GDI score, which is built on the angular deviation of projected hidden-state displacement vectors, can therefore be seen as a **forward-only proxy for output-layer gradient conflict** — a meaningful, if partial, theoretical grounding.
> > >
> > > We are careful to state that this argument applies most cleanly to the output projection layer and does not constitute a full proof for all layers of the network. Extending this reasoning to the full parameter space requires assumptions about how representation-space geometry propagates through the Jacobian of the network — an important and non-trivial theoretical challenge.
> > >
> > > **Re-framing the empirical evidence.** Given this partial theoretical picture, we would like to re-frame the KL divergence evidence from our previous rebuttal not merely as an "additional experiment," but as a **theoretically predicted consequence**. If GDI-based filtering reduces the directional conflict among gradient contributions, then the resulting mini-batch gradient estimator will have lower variance, and the policy parameters will update along a more consistent direction across iterations — which is precisely what a lower and more stable KL trajectory captures.
> > >
> > > ---
> > >
> > > **Our Commitment for the Further Version**
> > >
> > > Based on your valuable guidance, we commit to the following concrete revisions in the camera-ready paper:
> > >
> > > 1. **Expanding Sec. 3.3** to include the partial theoretical analysis above — clearly distinguishing what can be argued rigorously (output-layer gradient correspondence) from what remains an open problem.
> > > 2. **Re-framing the KL divergence results** as empirical validation of the theoretical prediction, strengthening the narrative link between the geometric signal and optimization-level behavior.
> > >
> > > ---
> > >
> > > We are grateful for your thorough engagement throughout this review process, and we believe the paper will be stronger for it. We hope this response addresses your remaining concern, and we welcome any further discussion.

---

### Official Review · Reviewer_pBjQ · 2026-03-12

**Soundness:** 3
**Presentation:** 3
**Significance:** 3
**Originality:** 4
**Overall Recommendation:** 4
**Confidence:** 4

**Summary:**

The paper point out that the instability of online reinforcement learning for LLMs stems not only from noisy rewards, but more fundamentally from the fact that "certain high-reward rollouts point toward update directions in the representation space that are inconsistent with the majority of samples", thereby derailing the training process. To address this, the paper proposes GEOALIGN, which performs a geometric consistency filtering and correction step before each RL update.

**Compliance With Llm Reviewing Policy:**

Affirmed.

**Key Questions For Authors:**

1. Could the evaluation be extended to at least 8B scale to better demonstrate the scalability of the proposed method?

2. In tasks such as math, where outcomes are programmatically verifiable, wouldn't these rollouts (high reward with different directions) be valuable or even represent breakthroughs, rather than being treated as harmful? If so, how could the proposed method avoid mistakenly
penalizing such beneficial high-reward rollouts?

3. Have the authors considered experimenting with other simple and commonly used architectures as projector?

**Limitations:**

Yes, the authors discuss their limitations, but I think this could be written as a standalone section.

**Strengths And Weaknesses:**

Strengths:
1. The paper offers a highly novel perspective. Conventionally, one would expect models to optimize toward high-reward directions, but the authors put forward the insight that high-reward samples with different direction may not beneficial and illustrate this point effectively.

2. The experimental section employs extensive visualizations and figures to demonstrate the effectiveness of the proposed method.

Weaknesses:
1. The experiments are conducted only on 1.7B and 4B models. I believe the evaluation could be extended to at least 8B scale (and larger if resources permit).

2. The paper's core thought is that high rewards can sometimes be harmful. However, in tasks such as mathematics, where outcomes are programmatically verifiable, high reward values typically indicate correct results. Such rollouts (with different direction) are often valuable and potentially represent breakthroughs, and should not be treated as harmful.

3. When selecting the optimal number of MLP layers, the authors could also experiment with other simple and commonly used architectures.

---

> ### Author Rebuttal · Authors · 2026-03-31
>
> Thank you for your insightful review and recognition of our novel geometric perspective and extensive experimental validation. We address each Weakness (W) and Question (Q) below. Regarding all Tables and Figures referenced in this rebuttal, we have included them in our [anonymous repo](https://anonymous.4open.science/r/GeoAlign-Rebuttal-E5F7/rebuttal_experiments.md).
>
> ---
>
> **W1 & Q1: On Scalability to Larger Models**
>
> > *"... Could the evaluation be extended to at least 8B scale?"*
>
> We have conducted additional experiments on **Qwen3-8B** HH-RLHF, with the same hyperparameters as the main submission. Due to the limitation of resources and time, we included only one experiment comparing raw and ours, results are shown in **Table R5 & Fig. R3** (anonymous repo).
>
> Crucially, analysis of the 8B training run confirms that the **long-tail GDI distribution** (the geometric signature of directional inconsistency) persists at this scale—the phenomenon is not an artifact of smaller models. The full training curves show consistently smoother optimization compared to BASE-GRPO. We will incorporate these 8B results into the final version.
>
> ---
>
> **W2 & Q2: On Preserving Valuable High-Reward "Breakthrough" Rollouts in Math**
>
> > *"In math tasks where outcomes are programmatically verifiable, wouldn't high-reward rollouts with different directions be valuable breakthroughs rather than harmful?"*
>
> We appreciate this important question. We clarify that **GeoAlign does not penalize valid diverse solutions**—it targets geometric outliers that are structurally different from diverse-but-correct rollouts.
>
> **Point 1: Valid diverse solutions jointly define the prototype, not conflict with it.**
> Multiple correct reasoning paths for the same prompt all appear as "winners" in within-prompt preference pairs. Their projected directions collectively contribute to the consensus prototype $v_{\text{proto}}$ (Eq. 5). Consequently, valid diverse solutions are *aligned with* the consensus manifold and are not flagged. Only rollouts whose directions are **simultaneously extreme angular outliers across many within-prompt comparisons** receive high GDI—and as shown by the long-tail distribution in **Fig. 3(b)**, only the most extreme angular outliers (<5%) are penalized.
>
> **Point 2: High verified reward ≠ correct reasoning process.**
> Programmatic verifiers check the final answer but not the reasoning chain, leaving them vulnerable to pseudo-correct rollouts—e.g., flawed exhaustive searches that accidentally reach the right answer, or shortcut logic that does not generalize. These rollouts carry high rewards yet induce representation directions that conflict with rigorous multi-step deduction, making them genuine directional outliers. Our GDI score identifies precisely this pattern.
>
> **Empirical evidence:** As shown in **Table R6** (anonymous repo), we tracked the proportion of reward=1 samples in the full batch vs. the GeoAlign-filtered subset across training on Math.
>
> The proportion of correct samples in the filtered subset **mirrors or exceeds** their natural growth rate—GeoAlign does not systematically remove high-reward rollouts. Furthermore, **Pass@16** results on AIME24/25 and AMC23 (**Fig. R2** in anonymous repo) show that GeoAlign maintains a Pass@k slope equal to or steeper than BASE-GRPO at all $k$, confirming that the model's capacity to discover diverse correct solutions is fully intact.
>
> ---
>
> **W3 & Q3: On Alternative Projector Architectures**
>
> > *"... the authors could also experiment with other simple and commonly used architectures."*
>
> We evaluated alternative projector architectures on the Math task under identical training budgets. Results are shown in **Table R7** (anonymous repo).
>
> Our **Linear + ReLU** design achieves the optimal accuracy-efficiency balance. Linear + Residual yields a marginal +0.2% gain at the cost of 17% more compute, while heavier architectures (Bottleneck, Transformer Block) incur substantial overhead with no accuracy benefit. This confirms that the geometric signal is intrinsically simple and does not require complex parameterization, consistent with the high projector validation accuracy reported in **Fig. 9** of the main paper.
>
> ---
>
> **Closing Remarks**
>
> We thank you again for the constructive and insightful feedback. In summary, we have provided (i) 8B-scale validation confirming the scalability of directional inconsistency and GeoAlign, (ii) a conceptual and empirical clarification that GeoAlign preserves valid diverse solutions while targeting pseudo-correct outliers, and (iii) a comprehensive projector architecture ablation. We hope these responses have addressed your concerns, and we warmly welcome any further questions.

---

> > ### Author Rebuttal · Reviewer_pBjQ · 2026-04-06
> >
> > The authors have well addressed most of my concerns. Considering the overall quality of this work, I would like to keep my original positive rating score.

---

> > > ### Author Response · Authors · 2026-04-08
> > >
> > > Thank you for your positive assessment and for carefully reviewing our rebuttal. We are very glad to hear that your concerns have been adequately addressed, and we sincerely appreciate your support in maintaining a positive rating for our work.
> > >
> > > Thank you again for your time and consideration!

---

### Official Review · Reviewer_cA5E · 2026-03-15

**Soundness:** 2
**Presentation:** 1
**Significance:** 3
**Originality:** 3
**Overall Recommendation:** 4
**Confidence:** 4

**Summary:**

This paper studies instability in online RL for LLM alignment and argues that the main issue is not only noisy scalar rewards, but also a geometric failure mode they call directional inconsistency: some high-reward rollouts induce latent preference directions that disagree with the batch majority and destabilize updates. The proposed method uses detached hidden states to build within-prompt preference directions, learns a small projector to reveal a concentrated improvement manifold, flags angular outliers relative to a batch consensus prototype, and rectifies those rollouts before the RL update.

**Compliance With Llm Reviewing Policy:**

Affirmed.

**Final Justification:**

My major concerns have been well addressed after the discussion.

**Key Questions For Authors:**

- It is unclear whether the replacement-based rectification still optimizes the original RL objective with respect to an arbitrary reward function $r$ in an unbiased way, so the optimization target needs sharper justification.
- see also weaknesses

**Limitations:**

yes

**Strengths And Weaknesses:**

## Strengths
- The paper presents a clear and intuitive geometric perspective on RL robustness, and the method is lightweight compared with alternatives.
- The evaluation spans two fairly different reward settings, two model sizes, and several recent robust-RL baselines, which helps the claim look broader than a single benchmark win.
- The corruption experiments and anomaly-amplify ablation provide useful evidence that high-GDI rollouts are not just unusual, but often harmful to training.

## Weaknesses
- The paper is motivated as addressing training instability, but it provides little direct instability analysis such as reward-curve variance, or oscillation statistics; that evidence would make the motivation much stronger.
- The paper seems to conflate instability with objective misalignment from noisy or misspecified rewards, and the current experiments more clearly support robustness to misspecified rewards than improved optimization stability per se.
- The method appears to rely on a single dominant improvement direction, which may be reasonable for tasks with one preferred mode but is less convincing when multiple equally valid or equally preferred solution modes exist.
- The paper says it will ablate replacement against zero-weighting and dropping, but that comparison is not clearly shown in Sec. 5.4.

---

> ### Author Rebuttal · Authors · 2026-03-31
>
> Thank you for your thoughtful review and recognition of our geometric perspective and empirical breadth. We address each Weakness (W) and Question (Q) below. We included Tables and Figures for rebuttal in this [anonymous repo](https://anonymous.4open.science/r/GeoAlign-Rebuttal-E5F7/rebuttal_experiments.md).
>
> ---
>
> **W1: On the Direct Evidence of Training Instability**
>
> > *"little direct instability analysis such as ..."*
>
> We provide two complementary instability metrics across both tasks in Table R1-R3 & Fig. R1 (anonymous link).
>
> **Oscillation (Volatility):** We define $\text{Oscillation} = \frac{1}{T-1}\sum_{t=2}^{T}|Acc_t - Acc_{t-1}|$ to measure the change between checkpoints. GeoAlign reduces volatility by **18.6%** (0.0102 → 0.0083) and narrows cross-seed Std. over the last 5 checkpoints (0.0061 → 0.0052).
>
> **PPO Optimization Conflict (Table R1):** We monitor `pg_clipfrac` (fraction of clipped policy-gradient updates) and KL divergence as direct indicators of optimization conflict.
>
> GeoAlign consistently reduces both metrics, confirming fewer extreme updates from directional outliers. The evaluation dynamics in **Fig. 4** and multi-seed analysis in **Appendix C.3** further corroborate this.
>
> ---
>
> **W2: On Instability vs. Reward Misspecification**
>
> > *"conflate instability with objective misalignment"*
>
> We clarify that objective misalignment (cause) and optimization instability (effect) are two sides of the same mechanism in our framework (Sec. 3.2–3.3): both reward noise and internal representation artifacts manifest as *directional inconsistency*, which directly inflates gradient variance (Sec. 3.3).
>
> Crucially, the **Math task** provides a clean dissociation—its rewards are derived from a deterministic verifier, minimizing misspecification. Yet GEOALIGN still yields significant gains (**Table 1**: +2.5% on 1.7B) and reduces pg_clipfrac (see **W1** above).
>
> ---
>
> **W3: On the Single-Direction Assumption**
>
> > *"... single dominant improvement direction; less convincing when multiple equally valid solution modes exist"*
>
> We clarify two points:
> 1. GeoAlign assumes a *shared improvement manifold*, not a single solution path. The consensus prototype $v_{\text{proto}}$ (Eq. 5) is the normalized centroid of all projected directions—a vote-aggregated direction that inherently accommodates diverse solution modes, where multiple correct solutions collectively define the reward-improving axis rather than competing with each other.
>
> 2. To directly test whether GeoAlign suppresses solution diversity, we report **Pass@k** ($$\text{Pass}@k = 1 - \frac{\binom{n-c}{k}}{\binom{n}{k}}$$) on the challenging benchmarks (AIME24, AIME25, AMC23):
>
> Results are shown in **Fig. R2** (anonymous repo). GeoAlign maintains a Pass@k slope equal to or steeper than BASE-GRPO across all $k$ values, suggesting that GeoAlign primarily intervenes on directional outliers (e.g., reward-hacking or contradictory logic chains), not valid alternative reasoning modes.
>
> ---
>
> **W4: On the Rectification Ablation**
>
> > *"... ablation against zero-weighting and dropping is not shown"*
>
> We provide the end-to-end ablation in **Table R4**. The results reveal a strict hierarchy (**Replacement > Zero-weighting > Dropping**), driven by how well each strategy preserves per-prompt group size $K$ and gradient density under GRPO's intra-prompt advantage normalization:
>
> - **Dropping (worst):** Reduces $K$, altering the advantage normalization denominator and injecting artificial variance.
> - **Zero-weighting (sub-optimal):** Preserves $K$ but passively neutralizes anomalous rollouts, reducing effective update density.
> - **Replacement (best):** Preserves both $K$ and full gradient density by substituting anomalies with stable within-prompt alternatives.
>
> ---
>
> **Q1: On the Unbiasedness of Replacement-Based Rectification**
>
> > *"unclear whether replacement-based rectification still optimizes the original RL objective in an unbiased way"*
>
> We acknowledge the formal bias, but argue it is a necessary trade-off for robust RL:
>
> *Adaptive In-distribution Resampling*: By replacing outliers with same-prompt samples, GEOALIGN performs adaptive resampling to prioritize high-consensus gradients, robustly approximating the original objective without OOD data.
>
> *Online Curation against Reward Artifacts*: Proxy rewards often contain artifacts. Rather than an unbiased estimation of noisy signals, GEOALIGN filters directional outliers to ensure geometric consistency, bypassing reward traps to converge toward true human preferences.
>
> Conclusion: GEOALIGN trades minimal formal bias for lower-variance, more reliable gradient estimates, which is more justifiable than pure unbiased optimization in noisy RL environments.
>
> ---
>
> **Closing Remarks**
>
> We thank you for the careful and constructive feedback. We sincerely hope these responses support a reconsideration of the score—any improvement in your evaluation would be a great encouragement to us. We warmly welcome any further questions.

---

> > ### Author Rebuttal · Reviewer_cA5E · 2026-04-05
> >
> > Thank you to the authors for the detailed rebuttal. While it helps clarify several points, I remain unconvinced on the following issues:
> >
> > - From the authors' narrative in rebuttal, misalignment is presented as a key factor contributing to optimization instability. But verifiable domains do not fall into the scope of the story. In presentation, it would be more precise to position the method as primarily targeting RLHF settings. The empirical effectiveness on verifiable domains can then be highlighted as a positive secondary result, rather than being tied to the same core explanation.
> > - W3 - To clarify my earlier point: in RLHF settings, a single prompt can admit multiple equally valid and preferred responses (e.g., A and B). In such cases, the preference landscape is inherently multi-modal, and there may exist multiple valid optimization directions. My concern is that the proposed method appears to bias learning toward a single dominant direction, potentially collapsing these modes.
> >    Importantly, an optimal multi-modal policy is not equivalent to an optimal single-mode policy, even if both exhibit surface-level diversity.

---

> > > ### Author Response · Authors · 2026-04-06
> > >
> > > Thank you for your continued engagement and for sharpening the two remaining concerns so precisely. We address each one directly below.
> > >
> > > ---
> > >
> > > **On the Narrative Framing of Verifiable Domains**
> > >
> > > We appreciate this suggestion and agree that the paper can be more precise in distinguishing the source of misalignment across reward settings.
> > >
> > > At the same time, we would like to explain why we believe a unified geometric narrative remains the more complete characterization of GeoAlign's effectiveness. Programmatic verifiers check only final-answer correctness, not reasoning quality, making them structurally vulnerable to false positives: a rollout with flawed reasoning that reaches the correct answer receives the same reward as a rigorous deduction. We observed this directly in our training runs. For a problem asking for all integers $k$ such that the roots of $f(x) = x^3 - (k-3)x^2 - 11x + (4k-8)$ are integers, a model obtained reward=1 via an incomplete exhaustive search while incorrectly claiming a unique solution existed. Its latent representation conflicted sharply with the dominant direction of rigorous deduction — a clear directional outlier that GeoAlign successfully identified and rectified.
> > >
> > > In this sense, both reward settings share the same underlying geometric instability mechanism; they differ only in the proximate source of misalignment: reward incompleteness in verifiable domains versus reward model uncertainty in RLHF. We will follow the reviewer's suggestion to make this distinction explicit in the presentation, while maintaining the unified narrative as a more complete account of when and why GeoAlign helps.
> > >
> > > ---
> > >
> > > **On W3: Multi-Modal Policy Collapse**
> > >
> > > We thank the reviewer for this precise and important distinction — and we agree that surface-level diversity is not equivalent to multi-modal policy optimality. We address the concern at the mechanistic and empirical levels in turn.
> > >
> > > **Mechanism: $v_{\text{proto}}$ carries no systematic bias among equally preferred modes.**
> > >
> > > We note one implementation detail that was omitted from the current draft and will be added in the revision: preference pairs are weighted by reward margin when constructing the consensus prototype:
> > >
> > > $$v\_{\text{proto}} = \text{norm}\left(\sum\_{(i,k,\ell)\in\mathcal{P}}(r\_{i,k}-r\_{i,\ell})\cdot v\_{i,k,\ell}\right)$$
> > >
> > > Suppose valid modes A, B, and C are all preferred over low-reward response F. The pairs A–F, B–F, and C–F carry large reward margins and dominate the prototype. The pairs A–B, A–C, and B–C carry small margins and contribute minimally; moreover, across prompts their directions are isotropic and cancel statistically. As a result, $v_{\text{proto}}$ captures only the shared "valid vs. invalid" signal — it carries **no systematic preference among valid modes**. Directional outlier detection therefore flags deviations from this consensus, not rollouts representing distinct-but-valid solution paths.
> > >
> > > **Per-step independence prevents cumulative mode suppression.**
> > >
> > > The consensus prototype is recomputed independently at each step from the current batch, with no cross-step memory. Only the top-5% of rollouts satisfying a dual-threshold criterion are ever rectified. For a valid mode to be systematically suppressed, it would need to consistently occupy this extreme anomaly region across steps — which the margin-weighted construction above makes mechanistically implausible.
> > >
> > > **Empirical evidence targeting the reviewer's specific concern.**
> > >
> > > We take seriously the reviewer's point that surface-level diversity does not guarantee multi-modal optimality. To target this more directly, for all HH-RLHF test prompts we sample 32 responses per prompt and retain the top-8 by ArmoRM score, constructing a **high-reward subset** $\mathcal{Y}^+$. If GeoAlign had collapsed multiple valid modes into a single dominant pattern, probability mass within $\mathcal{Y}^+$ would concentrate on a narrow region of the representation space. We measure diversity within this high-reward subset at two levels — textual and semantic:
> > >
> > > $$\text{div}\_{\text{text}} = 1 - \frac{1}{|\mathcal{Y}^+|^2}\sum\_{i,j}\text{ROUGE}(y_i,y_j), \qquad \text{div}\_{\text{hidden}} = \frac{1}{|\mathcal{Y}^+|^2}\sum\_{i,j}(1-\cos(h_i,h_j))$$
> > >
> > > | Model | div\_text | div\_hidden |
> > > |---|---|---|
> > > | BASE-GRPO | 0.4932 | 0.1077 |
> > > | GeoAlign | 0.5508 (+11.7%) | 0.1076 (≈) |
> > >
> > > GeoAlign achieves higher textual diversity while occupying a near-identical semantic volume — consistent with probability mass remaining distributed across multiple valid modes rather than collapsing onto a single dominant pattern. This is the distinction the reviewer draws: an optimal multi-modal policy preserves coverage across valid modes, and GeoAlign does so at least as well as BASE-GRPO, if not better.
> > >
> > > ---
> > >
> > > We hope these responses address the remaining concerns. We remain open to further discussion and are committed to incorporating the narrative clarification into the final version.

---

### Decision · Program_Chairs · 2026-04-30

**Decision:**

Accept (regular)

**Comment:**

This paper presents a novel and intuitive geometric perspective on instability in online LLM RL and introduces a lightweight, forward-pass-only rollout curation method with modest but consistent gains across dialogue alignment and mathematical reasoning, including robustness under reward corruption. The main concerns were the lack of stronger theoretical grounding, larger-scale validation, and possible multi-modal preference settings, but the rebuttal addressed most of these points with additional analyses and 8B results, leaving limitations that can be clarified in the final version.